# Adaptive Data Augmentation on Temporal Graphs

**Yiwei Wang**[1]    **Yujun Cai**[2]    **Yuxuan Liang**[1]    **Henghui Ding**[3]
**Changhu Wang**[3]    **Siddharth Bhatia**[1]    **Bryan Hooi**[1]
[1] National University of Singapore
[2] Nanyang Technological University
[3] ByteDance
wangyw_seu@foxmail.com, {yujun001,ding0093}@e.ntu.edu.sg,
yuxliang@outlook.com, changhu.wang@gmail.com,
{siddharth,bhooi}@comp.nus.edu.sg

## Abstract

Temporal Graph Networks (TGNs) are powerful on modeling temporal graph data based on their increased complexity. Higher complexity carries with it a higher risk of overfitting, which makes TGNs capture random noise instead of essential semantic information. To address this issue, our idea is to transform the temporal graphs using data augmentation (DA) with adaptive magnitudes, so as to effectively augment the input features and preserve the essential semantic information. Based on this idea, we present the **MeTA** (*Memory Tower Augmentation*) module: a multi-level module that processes the augmented graphs of different magnitudes on separate levels, and performs message passing across levels to provide adaptively augmented inputs for every prediction. MeTA can be flexibly applied to the training of popular TGNs to improve their effectiveness without increasing their time complexity. To complement MeTA, we propose three DA strategies to realistically model noise by modifying both the temporal and topological features. Empirical results on standard datasets show that MeTA yields significant gains for the popular TGN models on edge prediction and node classification in an efficient manner.

## 1   Introduction

Many real-world graphs are not static but evolving, where every edge (or interaction) has a timestamp to denote its occurrence time. These graphs are called temporal (or dynamic) graphs [39]. Recently, temporal graph networks (TGNs) [25, 39, 16] have been proposed to support learning on temporal graphs. Advanced TGNs utilize an RNN based memory module to represent a node's history as a compact state (see Fig. 1), which is used to predict the node's activities [28]. TGNs are capable of making predictions from complex graph topology and temporal information , thanks to their advanced representational power. However, the increased representational capacity comes with higher model complexity, which can induce over-fitting and weaken their generalization ability. In particular, a trained TGN may capture random noise instead of semantic information, which is not desired [41].

To combat over-fitting, data augmentation (DA) has been demonstrated to be effective [23]. Nevertheless, DA for the temporal graphs remains under-explored, of which the main challenges lie in highly irregular dynamic topology. DA applies transformations to input features so as to model realistic noise for enriching the input data. The magnitudes of the transformations, known as *DA magnitudes*, are controlled by hyper-parameters, which is positively related to the difference between the input features before and after DA [5]. Existing work on image and text data devises *adaptive DA* methods, which apply higher DA magnitudes to the less informative parts of input features, in order to effectively augment the input features while preserving the essential semantic information [38].

35th Conference on Neural Information Processing Systems (NeurIPS 2021).

In order to design such an adaptive DA method on temporal graphs, we consider the informativeness of edges to a target node in terms of both time and topology when predicting the activities of the target node. On the time axis, more recent edges tend to be more informative for predicting the target node's states than the earlier ones. For example, when a girl is going to watch a movie in the evening, the interaction between her and a cinema conductor a few minutes ago is much more informative to predict her behaviors than the academic discussion between her and a student in the morning. Similarly, in terms of topology, the edges directly connecting the target node provide more important information than edges a few hops away. Overall, the edges that are closer to the target node in time or topology tend to provide more important information, as validated in the existing work [25].

To devise an efficient adaptive DA method for temporal graphs, the central idea of this paper is to generate a few graphs with different DA magnitudes, and perform the message passing between these graphs to provide adaptively augmented inputs for every prediction. We encapsulate this idea in a novel module, called **MeTA** (*Memory Tower Augmentation*): MeTA stacks a few levels of memory modules as a tower with weight sharing, where lower levels process the graphs of lower DA magnitudes (see Fig. 2). MeTA includes two message passing mechanisms across levels for adaptive DA. The first one is *cross-level propagation*, which propagates the features between nodes across levels, while the second one is *memory transition*, that transmits memory states of a node from higher levels to lower ones periodically. This design allows us to achieve the goal of adaptive DA, since for every prediction, the edges closer to the target node in time or topology are placed on lower levels corresponding to lower magnitudes, as demonstrated by the theoretical analysis (see Sec. 3.4).

To complement MeTA, we propose three DA strategies augmenting both the temporal and topology features: (i) perturbing the edge time to simulate time shifts, (ii) removing edges, and (iii) adding edges to modify the topology. Our strategies effectively enrich the input data by modeling realistic noise intuitively and theoretically. MeTA is a general module that can be applied for training popular TGNs to enhance their performance. MeTA improves the effectiveness of TGNs without increasing their time complexity during training, as analyzed in Sec. 3.4. Note that our methods do not induce any extra inference cost since DA is not applied during inference.

We evaluate our MeTA on edge prediction and node classification tasks using the standard temporal graph datasets: Reddit [3], Wikipedia [20], MOOC [16]. We measure its performance through the metrics: test accuracy, average precision (AP), and the area under the ROC accuracy curve (AUC), under inductive and transductive settings. Overall, MeTA achieves substantial improvements for popular TGN models [25, 39] and enhances them to outperform the baseline methods.

## 2   Related Work

Although many methods have been proposed for representation learning on static graphs [24, 9, 14, 15, 10, 29, 27, 18, 32, 34, 36], the work on representation learning in temporal graphs is much sparser. There exist two main classes of temporal graphs: discrete-time dynamic graphs (DTDG) [17] and continuous-time dynamic graphs (CTDG) [25]. DTDG are sequences of static graph snapshots taken at intervals in time, while CTDG can be represented as timed lists of events. Representation learning on CTDG is more flexible, general and challenging, which is the focus of this paper. Early models for temporal graph learning focus on DTDGs [11, 17, 1, 8, 40]. Some recent work addresses the learning on CTDGs [16, 28, 25, 39, 30] based on the increased model complexity. For example, [31] proposes a temporal graph learning model based on causal anonymous walks, which uses RNNs to encode the temporal information. Our method is orthogonal to these work in the sense that we do not propose a new TGN model, but acts as a general DA module that can be flexibly incorporated into the training of popular TGNs to improve their effectiveness.

Data augmentation plays a central role in training neural networks, of which the effectiveness has been validated on image data [42, 26, 7]. On the data augmentation for the static graph data, [35] proposes NodeAug to augment and utilize the unlabeled data in semi-supervised learning, and [33] crops the subgraphs to augment the input features for the static graph classification. In [37], the authors propose the Mixup methods for static graphs, which interpolate the input features of both nodes and graphs in the semantic space as data augmentation. Different from them, our work focuses on the DA for temporal graphs. Our MeTA considers the edge importance related to the temporal and topology information and adaptively augment temporal graphs in an efficient manner, while our DA strategies effectively augment both the temporal and topological features.

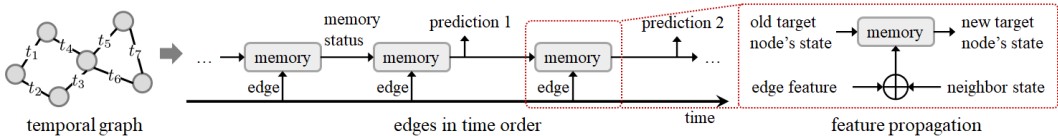

Figure 1: TGNs feed edges in time order into an RNN-based memory module, which updates nodes' states for predictions. Every time an edge connecting the node $i$ happens, the memory module takes the $i$'s state, edge feature, and the connected neighbor's state as the inputs, and update the state of node $i$ (see Eq. (1)). In this way, the memory state of node $i$ represents $i$'s history.

## 3  Methodology

In this section, we propose MeTA (*Memory Tower Augmentation*), an adaptive data augmentation (DA) approach for improving the temporal graph learning. A MeTA module can be applied to popular TGN models, and trains them with DA to improve its inference performance. MeTA includes two message passing mechanisms: *cross-level propagation* and *memory transition*, to provide adaptively augmented inputs for every prediction, as demonstrated by the theoretical analysis.

To complement MeTA, we propose three DA strategies for augmenting temporal graphs by modifying the temporal and topological features, which effectively model the realistic noise to enrich the input data intuitively and theoretically. Using our methods to enhance the popular TGNs does not increase their time complexity and does not induce any inference cost.

### 3.1  Preliminaries

In a temporal graph $\mathcal{G}$, an interaction between nodes $i$ and $j$ is denoted as a temporal edge $\mathbf{e}_{ij}(t), t > 0$, where $\mathbf{e}$ is the edge attributes [25], and $t$ is the edge time. $t = 0$ is generally the time of starting observations.

Advanced TGN models utilize a memory module to store and update the (memory) state $\mathbf{s}_i(t)$ of node $i$ (see Fig. 1). The state of node $i$ is expected to represent $i$'s history in a compressed format. Given the memory updater as mem, when an edge $\mathbf{e}_{ij}(t)$ connecting node $i$ is observed, node $i$'s state is updated as:

$$\mathbf{s}_i(t) = \mathsf{mem}\big(\mathbf{s}_i(t^-), \mathbf{e}_{ij}(t)\|\mathbf{s}_j(t^-)\big), \tag{1}$$

where $\mathbf{s}_i(t^-)$ is the memory state of node $i$ just before time $t$, $\|$ is the concatenation operator, and node $j$ is $i$'s neighbor connected by $\mathbf{e}_{ij}(t)$. $\mathbf{s}_i(t)$ is initialized to 0 values and not changed until the next edge involving node $i$ happens. Edge feature $\mathbf{e}_{ij}(t)$ and neighbor state $\mathbf{s}_j(t^-)$ are concatenated as the input to update $\mathbf{s}_i(t)$. All edges are fed to the memory modules in time order (see Fig. 1).

The implementation of mem varies slightly for different TGN models, of which a common component is a recurrent neural network (RNN) such as LSTM [12] or GRU [4]. TGNs make the prediction for node $i$ at time $t$ using $\mathbf{s}_i(t^-)$, instead of $\mathbf{s}_i(t)$, since when making predictions for time $t$, the interactions at time $t$ have not happened and cannot be observed.

The number of hops is used to denote the distance on the topology [22]. From Eq. (1), we observe that the edges connecting node $i$ directly contributes to $i$'s state, while the edges, that are multiple hops away from $i$, have their features propagated to node $i$ through $i$'s neighbors' states.

### 3.2  Memory Tower for Adaptive Data Augmentation on Temporal Graphs

Data augmentation (DA) aims at creating novel training data by applying transformation to input features [38]. The magnitudes of the transformations, known as *DA magnitudes*, are controlled by the hyper-parameters, which is positively related to the difference between the input features before and after DA [5]. To effectively augment the input features while preserving the essential semantic information, *adaptive DA* methods apply low DA magnitudes to the informative parts of the input features and higher magnitudes to the less informative parts [38]. For example, TF-IDF based word replacement, a DA method for natural language processing, is designed to remove less important

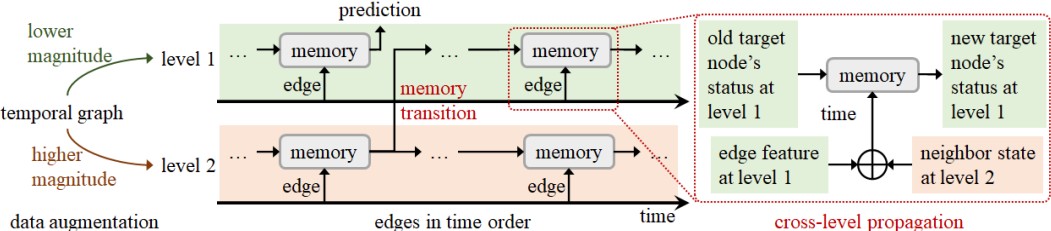

Figure 2: Our MeTA method with two levels of memory modules. The colors denote the DA magnitudes, i.e., 'green' means lower, and 'yellow' means higher. With the memory transition from level 2 to level 1, for every prediction, the earlier edges hold higher DA magnitudes. By propagating the neighbors' states from higher levels, more distant edges on topology hold higher DA magnitudes.

keywords with higher probability [38]. This helps the model to generalize better to changes in the uninformative features, and effectively encode the essential features into the representations.

In a temporal graph, when predicting the activities of a target node, different edges are of different informativeness for the target node. For example, an engineer is busy doing a project in the evening, the interaction between him and his colleague a few minutes ago is much more important for predicting his working status than the discussion between him and his daughter on a new movie in the morning. Similarly, the edges directly connecting the target node tend to be more informative than the ones that are multiple hops distant since the former directly happen on the target node. Overall, if an edge is more close to the target node in terms of time or topology, it is more informative, as validated in [25].

Applying adaptive DA magnitudes to edges of different informativeness is not trivial, since an edge's informativeness changes with different predictions, i.e., when making predictions for different target nodes, the distance from an edge to the target node changes. A simple method for adaptive DA is to separately augment the graph with adaptive magnitudes for every prediction, which leads to different input graphs for different predictions. Therefore, this method repeats feeding the graph edges to the model from the beginning time for every prediction, and is named the *refeed* method.

Existing work [25, 28, 16] trains the memory module in a recursive manner for efficiency, i.e., the predictions at time $t$ take the former output at time $\tau$ ($\forall \tau < t$) as inputs, as shown in Fig. 1, since there is no DA and different predictions share the same input graph. The refeed method is much less efficient than recursive training, since in the former one, the memory states before the prediction time have to be recomputed for every prediction, while the number of predictions is generally the same as the number of edges during training, which is a large value in practice [16].

We propose the Memory Tower Augmentation (MeTA), which offers appropriate DA magnitudes for every prediction in an efficient manner. MeTA stacks $L$ levels of memory modules as a tower with weight sharing. For level $l$ ($l = 1, \ldots, L$), we augment the graph $\mathcal{G}$ as $\mathcal{G}^{(l)}$ with the magnitude $p_l$. $p_l$ increases as $l$ increases, so that the DA magnitude is higher at a higher level. Denote the state of node $i$ at time $t$ on level $l$ as $\mathbf{s}^{(l)}(t)$, and the edge connecting nodes $i$ and $j$ at time $t$ on level $l$, i.e., belonging to $\mathcal{G}^{(l)}$, as $\mathbf{e}_{ij}^{(l)}(t)$. We feed edges at all levels in time order to the memory module, and always make the prediction using the memory states at the first level, i.e., $\mathbf{s}_i^{(1)}(t)$.

MeTA includes two message passing mechanisms across levels to offer adaptive DA for every prediction: *cross-level propagation* and *memory transition*. With *cross-level propagation*, when observing an edge $\mathbf{e}_{ij}^{(l)}(t)$ involving node $i$ at level $l$, node $i$'s state at level $l$ is updated as:

$$\mathbf{s}_i^{(l)}(t) = \mathsf{mem}\big(\mathbf{s}_i^{(l)}(t^-), \mathbf{e}_{ij}^{(l)}(t) \| \mathbf{s}_j^{(\min(l+1,L))}(t^-)\big), \quad (2)$$

In this way, the edges that are more distant to the target node $i$ in terms of the topology are augmented at a higher magnitude, since their features are propagated to the target node through the neighbors' states $\mathbf{s}_j^{(\min(l+1,L))}(t^-)$ that are located on higher levels (see Proposition 1).

In addition to the distances on the topology, the time distances also influence the edge's informativeness. Hence, the second message passing mechanism we design is the *memory transition* for nodes'

---

**Algorithm 1** Memory Tower for Adaptive Data Augmentation on Temporal Graphs

---

**Input:** Original Temporal Graph $\mathcal{G}$ with the node set $\mathcal{V}$ and the edge sequence $\mathbf{E}$ in time order with timestamps: $t_1, \ldots, t_N$ $(0 \leq t_1 \leq \cdots \leq t_N)$, where $N$ is the number of edges, a TGN model with the memory module mem, hyper-parameters $L$ for the level number , $T$ for the memory transition period, and the augmented graphs at different levels $\mathcal{G}^{(l)}, l = 1, \ldots, L$.

**Output:** The predictions made by the TGN model on time $t_1, \ldots, t_N$: $\{p(t_i)|i = 1, \ldots, N\}$

---

1: Initialize states $\mathbf{s}_i^{(l)}(0), \forall l = 1, \ldots, L, \forall i \in \mathcal{V}$, counter values $c_i, \forall i \in \mathcal{V}$, starting time $t_0$ to 0.
2: **for** $n \leftarrow 1$ to $N$ **do**
3:     $\mathbf{e}_{ij}(t_n) \leftarrow \mathbf{E}[n]$ (the $n$th element in $\mathbf{E}$)
4:     We order the edges from $\mathcal{G}^{(l)}, l = 1, \ldots, L$ between $t_{n-1}$ (inclusive) and $t_n$ as a sequence $\mathbf{E}_n$ with timestamps $\tau_1, \ldots, \tau_{N_n}(t_{n-1} \leq \tau_1 \leq \cdots \leq \tau_{N_n} < t_n)$.
5:     **for** $n' \leftarrow 1$ to $N_n$ **do**
6:         $\mathbf{e}_{km}^{(l)}(\tau_{n'}) \leftarrow \mathbf{E}_n[n']$ (the $n'$th element in $\mathbf{E}_n$)
7:         $\mathbf{s}_k^{(l)}(\tau_{n'}) \leftarrow \text{mem}\big(\mathbf{s}_k^{(l)}(\tau_{n'}), \mathbf{e}_{km}^{(l)}(\tau_{n'}) \| \mathbf{s}_m^{(\min(l+1,L))}(\tau_{n'}^-)\big)$
8:         $\mathbf{s}_m^{(l)}(\tau_{n'}^-) \leftarrow \text{mem}\big(\mathbf{s}_m^{(l)}(\tau_{n'}^-), \mathbf{e}_{km}^{(l)}(\tau_{n'}) \| \mathbf{s}_k^{(\min(l+1,L))}(\tau_{n'}^-)\big)$
9:     **end for**
10:     Make prediction $p(t_n)$ related to edge $\mathbf{e}_{ij}(t_n)$ through states on time $t_n^-$ at level 1.
11:     **for** $o$ in $\{i, j\}$ **do**
12:         $c_o \leftarrow c_o + 1$
13:         **if** $c_o \geq T$ **then**
14:             **for** $l \leftarrow 1$ to $L - 1$ **do**
15:                 $\mathbf{s}_o^{(l)}(t_n^-) \leftarrow \mathbf{s}_o^{(l+1)}(t_n^-)$
16:             **end for**
17:             $c_o \leftarrow 0$
18:         **end if**
19:     **end for**
20: **end for**

---

states from higher levels to lower ones, i.e., replacing the level-$l$ states with the level-$l + 1$ ones:

$$\mathbf{s}_i^{(l)}(t) \leftarrow \mathbf{s}_i^{(l+1)}(t), \forall l = 1, \ldots, L - 1. \tag{3}$$

The transition is made in the memory tower periodically, so that when we make a prediction at level 1, the more recent edges contributing to the prediction are at lower levels corresponding to lower DA magnitudes (see Proposition 2). We determine the transition time of node $i$ by implementing a counter for node $i$, which is initialized to 0 and increments when the model makes a prediction related to node $i$. If the counter reaches $T$, we reinitialize the counter and perform this memory transition for node $i$, where $T$ is a hyper-parameter for the transition period.

Overall, we visualize our MeTA method in Fig. 2 and summarize it in Alg. 1 (see Appendix). How to obtain the predictions from memory states and calculate the training loss differ slightly for the specific TGN models and downstream tasks, of which the details can be found in the corresponding work [16, 28, 25].

### 3.3 Data Augmentation Strategies

We propose three DA strategies for the temporal graphs: (i) perturbing time; (ii) removing edges; (iii) adding edges with perturbed time. The first one augments the temporal features; the second is for the graph topology, while the last augments both the topology and temporal features. The search space for the hyper-parameters of DA magnitudes is large if we set the parameters separately for different DA techniques. Hence, for the convenience of setting the hyper-parameters, we use a unified hyper-parameter $p$ to control the magnitude of all the three DA strategies. A higher value of $p$ implies a larger magnitude. Next, we introduce our DA strategies in detail.

**Perturb Time** Each edge has a timestamp to denote its happening time. In practice, these timestamps may not be precise. For instance, a courier can report delivery time earlier or later than the actual time. These time shifts are widespread in the real world but do not necessarily affect the semantic

information. Hence, we perform time perturbation to model this kind of noise as data augmentation. Denote the average intervals between interactions as $\bar{t}$, and the time range of training data as $[0, T_{\max}]$. We add Gaussian noise following $\tau \sim \mathcal{N}(0, \sigma^2)$ [2] to the edge time $t$ as $t_{\mathsf{new}} = \tau + t$, where $t_{\mathsf{new}}$ is the augmented timestamp. We set the standard deviation of Gaussian noise $\sigma$ as $10p\bar{t}$. When $p$ is 0, all edges' time is not changed, and the noise's standard deviation grows as $p$ increases. The augmented time $t_{\mathsf{new}}$ can exceed the time range $[0, T_{\max}]$. This is not practical since only the observations in the range $[0, T_{\max}]$ can be observed. Therefore, we set $t_{\mathsf{new}} = t$ if $\tau + t > T_{\max}$ or $\tau + t < 0$, so that all the augmented time $t_{\mathsf{new}}$ all in the valid range $[0, T_{\max}]$. This design also helps us to retain the original distribution of edges' happening time after DA (see Theorem 1).

**Remove Edges** Removing edges models the realistic noise that some edges may be missing in the data collection. For example, a researcher can communicate with hundreds of other researchers one year. Collection of these interactions may have omissions, and it is unnecessary to consider all interactions to predict the researcher's state. A satisfactory TGN model is desired to effectively encode the essential semantic information even in the presence of this noise. We set the probability of removing an edge as $p$. With a higher $p$, edges are more likely to be removed and the augmented graphs tend to be more incomplete.

**Add Edges with Perturbed Time** A node can repeat interacting with its neighbor. An engineer, for example, repeats communicating with his colleague to check the project progress, and he talks with a waiter several times to order more food during dinner. We add the edges with the perturbed time to model noise from interaction repetition. Denote the edge set of the original graph $\mathcal{G}$ as $\mathcal{E}$. We uniformly sample the edges from $\mathcal{E}$ with replacement with the budget $p\#\mathcal{E}$ to form $\hat{\mathcal{E}}$ and then perturb the time of edges in $\hat{\mathcal{E}}$ as introduced earlier , where $\#\mathcal{E}$ is the cardinality of $\mathcal{E}$. We add the edges in $\hat{\mathcal{E}}$ for DA. Higher $p$ corresponds to more edges being added.

## 3.4 Analysis and Discussion

We analyze how our MeTA and DA techniques work for temporal graph learning. Recall that if an edge is more distant to a target node on either topology or time, it is less important or informative to predict the target node's properties and thus we are expected to apply a higher DA magnitude to it. Next, we provide theoretical analysis to show that our methods meet this expectation. The proof is provided in Appendix.

**Proposition 1** (Effect of topology). *If edge $\mathbf{e}_{km}^{(l)}(\tau)$ contributes to the node state $\mathbf{s}_i^{(1)}(t)$, and the distances from nodes $k$ and $m$ to $i$ are equal to or larger than $h$ hops in graph $\mathcal{G}$, we have the level of edge $\mathbf{e}_{km}^{(l)}(\tau)$: $l \geq \min(h+1, L)$ and the time $\tau < t$, where $L$ is the number of levels.*

Proposition 1 implies that the edges more distant to the target node on the topology must be on levels of higher $l$, and thus of higher DA magnitudes. In addition to the topology, we provide the following proposition to illustrate the effects of the temporal distances.

**Proposition 2** (Effect of time). *If edges $\mathbf{e}_{ij}^{(l_1)}(\tau_1)$ and $\mathbf{e}_{ik}^{(l_2)}(\tau_2)$ connect node $i$ and directly contribute to the node state $\mathbf{s}_i(t)$ as Eq. (2), we have $l_1 \geq l_2, \tau_1 < \tau_2 < t$ when $\tau_1 < \tau_2$ holds, and $l_1 \leq l_2, \tau_2 < \tau_1 < t$ holds with $\tau_1 > \tau_2$.*

Proposition 2 shows that the earlier edges tend to be on levels of higher $l$, and thus of higher DA magnitudes.

Next, we analyze how our DA strategies retain the original temporal graph characteristics to model realistic noise. Many social and natural activities follow the Poisson process [19, 13, 6]. Denote the time range of the training data as $[0, T_{\max}]$. Our data augmentation includes the perturbations on the edge time. If nodes' interactions follows a Poisson process, we provide the following theorem to show that our DA techniques do not change the distribution of edge time.

**Theorem 1.** *If the edges before data augmentation follow a homogeneous Poisson process, our data augmentation does not change the distribution of any edge's occurrence time.*

Theorem 1 illustrates the benefits of our strategies for time perturbation. Otherwise, if the original distribution is broken, the augmented edges cannot reflect the realistic condition and may degrade the TGNs' generalization. In addition to the time distribution, we provide the following proposition to

show the expected edge number does not change after our DA. Denote the edge set of the original graph $\mathcal{G}$ as $\mathcal{E}$ and that of the augmented graph $\mathcal{G}^{(l)}$ as $\mathcal{E}^{(l)}$.

**Proposition 3.** *After our data augmentation, the expected edge number is the same as that before data augmentation for any data augmentation magnitude, i.e., $\mathbb{E}[\#\mathcal{E}^{(l)}] = \#\mathcal{E}, \forall p_l \in [0,1]$ holds.*

This proposition shows that our data augmentation does not change the density of interactions in expectation, which meets the realistic condition and does not induce extra computation load.

We analyze the time complexity of training TGNs and MeTA with respect to the input data. Denote the number of interactions as $\#\mathcal{E}$, the node number $\#\mathcal{V}$, and the dimensionality of the input features as $d$. Since the RNN based memory module updates memory states with every edge and make predictions in a recursive manner, the time complexity of the memory module is $\mathcal{O}(d \cdot \#\mathcal{E})$. The refeed method introduced in Sec. 3.1 achieves adaptive DA by feeding the input data from the beginning time to the prediction time into the memory module for every prediction. The corresponding time complexity is:

$$\mathcal{O}(\sum_{e=0}^{\#\mathcal{E}-1} de) = \mathcal{O}\big(d((\#\mathcal{E})^2 - \#\mathcal{E})/2\big) = \mathcal{O}\big(d(\#\mathcal{E})^2\big), \tag{4}$$

which is much higher than the complexity of original TGNs.

Our MeTA method stacks $L$ levels of memory modules. On level $l$, the time complexity is $\mathcal{O}(d \cdot \#\mathcal{E}^{(l)})$, where $\mathcal{E}^{(l)}$ is the edge set of the augmented graph $\mathcal{G}^{(l)}$ on level $l$. Summing the complexity of all levels up leads to $\sum_{l=1}^{L} d \cdot \#\mathcal{E}^{(l)}$, of which the expectation is

$$\mathcal{O}\Big(\mathbb{E}\big[\sum_{l=1}^{L} d \cdot \#\mathcal{E}^{(l)}\big]\Big) = \mathcal{O}\Big(\sum_{l=1}^{L} d \cdot \mathbb{E}[\#\mathcal{E}^{(l)}]\Big) = \mathcal{O}\Big(\sum_{l=1}^{L} d \cdot \#\mathcal{E}\Big) = \mathcal{O}(d\#\mathcal{E}), \tag{5}$$

which is same as the original TGNs. The second equation holds because of Proposition 3, while the last equation holds because $L$ is a small constant value invariant to the input data, and we find that $L = 2$ generally exhibits satisfactory performance. Compared to the refeed method, our MeTA achieves the adaptive DA with much lower complexity during training. During inference, our MeTA method does not lead to any extra computation load since DA is not applied then.

## 4 Experiments

In this section, we present the performance of TGN models implemented with our adaptive DA method. We compare our method against a variety of strong baselines on the task of edge prediction and node classification on temporal graphs. Our experimental settings closely follow those of the previous work [25, 39, 30] to ensure fair comparison. To avoid violating temporal constraints, we make predictions that strictly take place after all observations. As for the evaluation metrics, we follow [39, 30] to use average precision (AP) and accuracy in the edge prediction tasks and employ the area under the ROC accuracy curve (AUC) for node classification [16].

We use three standard temporal graph datasets: MOOC [16], Reddit [3], and Wikipedia [16] for evaluation. The statistics of these datasets are shown in Table 1. We use a chronological train-validation-test split with a ratio of 70%-15%-15% following [25, 39].

For the hyper-parameters of baseline methods, e.g., the number of hidden units, the optimizer, the number of sampled neighbors, and the learning rate, we set them as suggested by their authors. For the hyper-parameters of our MeTA, we set the number of memory levels as $L = 2$, the data augmentation magnitudes as $p_1 = 0.1, p_2 = 0.8$, and the memory transition period as $T = 10$, by default. Note that these settings are fixed for all experiments unless specifically indicated to be changed. Setting more levels with $L > 2$ offers higher granularity for DA magnitudes, but we find that $L = 2$ generally offers satisfactory performance.

### 4.1 Edge Prediction

Following [39, 25], we conduct experiments for edge prediction under both transductive and inductive settings for a comprehensive evaluation. In the former one, we predict edges connecting the nodes observed during training, and in the latter we predict edges for the unseen nodes [25].

Table 1: Statistics of the datasets used in our experiments. Edges refer to temporal edges.

| Dataset | #Nodes | #Edges | #Nodes in val./test. | #Edges in val./test. | Nodes' label type |
|---|---|---|---|---|---|
| MOOC | 7,144 | 411,749 | 2,599/2,412 | 61,762/61,763 | course dropout |
| Reddit | 10,984 | 672,447 | 9,839/9,615 | 100,867/100,867 | posting ban |
| Wikipedia | 9,227 | 157,474 | 3,256/3,564 | 23,621/23,621 | editing ban |

Table 2: Test accuracy and average precision (AP) of transductive edge prediction. We conduct 100 trials with random weight initialization. Mean (%) and standard deviations are reported. The best results in each column are highlighted in **bold** font.

| Method | MOOC | | Reddit | | Wikipedia | |
|---|---|---|---|---|---|---|
| | Accuracy | AP | Accuracy | AP | Accuracy | AP |
| CTDNE [21] | $65.34 \pm 0.7$ | $74.29 \pm 0.6$ | $73.76 \pm 0.5$ | $91.41 \pm 0.3$ | $79.42 \pm 0.4$ | $92.17 \pm 0.5$ |
| JODIE [16] | $76.45 \pm 0.6$ | $83.87 \pm 0.4$ | $90.91 \pm 0.3$ | $97.11 \pm 0.3$ | $87.04 \pm 0.4$ | $94.62 \pm 0.5$ |
| TGAT [39] | $75.20 \pm 0.5$ | $82.66 \pm 0.4$ | $92.92 \pm 0.3$ | $98.12 \pm 0.2$ | $88.14 \pm 0.2$ | $95.34 \pm 0.1$ |
| DyRep [28] | $73.36 \pm 0.4$ | $81.75 \pm 0.3$ | $92.11 \pm 0.2$ | $97.98 \pm 0.1$ | $87.77 \pm 0.2$ | $94.59 \pm 0.2$ |
| TGN [25] | $81.38 \pm 0.6$ | $89.79 \pm 0.5$ | $92.56 \pm 0.2$ | $98.70 \pm 0.1$ | $89.51 \pm 0.4$ | $98.46 \pm 0.1$ |
| DyRep + MeTA (Ours) | $76.21 \pm 0.4$ | $84.18 \pm 0.3$ | $93.04 \pm 0.3$ | $98.62 \pm 0.1$ | $88.92 \pm 0.2$ | $95.63 \pm 0.2$ |
| TGN + MeTA (Ours) | $\mathbf{83.84 \pm 0.5}$ | $\mathbf{92.03 \pm 0.3}$ | $\mathbf{94.19 \pm 0.2}$ | $\mathbf{99.08 \pm 0.1}$ | $\mathbf{91.34 \pm 0.3}$ | $\mathbf{98.87 \pm 0.1}$ |

Table 3: Test accuracy and average precision (AP) of inductive edge prediction. We conduct 100 trials with random weight initialization. Mean (%) and standard deviations are reported. The best results in each column are highlighted in **bold** font.

| Method | MOOC | | Reddit | | Wikipedia | |
|---|---|---|---|---|---|---|
| | Accuracy | AP | Accuracy | AP | Accuracy | AP |
| JODIE [16] | $75.79 \pm 0.5$ | $83.44 \pm 0.6$ | $88.34 \pm 0.9$ | $94.36 \pm 1.1$ | $84.32 \pm 0.4$ | $93.11 \pm 0.4$ |
| TGAT [39] | $74.02 \pm 0.3$ | $80.84 \pm 0.5$ | $90.73 \pm 0.2$ | $96.62 \pm 0.3$ | $85.35 \pm 0.2$ | $93.99 \pm 0.3$ |
| DyRep [28] | $72.92 \pm 0.4$ | $80.36 \pm 0.4$ | $89.60 \pm 0.2$ | $95.68 \pm 0.2$ | $83.46 \pm 0.3$ | $92.05 \pm 0.3$ |
| TGN [25] | $80.73 \pm 0.2$ | $89.21 \pm 0.3$ | $91.62 \pm 0.1$ | $97.55 \pm 0.1$ | $88.60 \pm 0.2$ | $97.81 \pm 0.1$ |
| DyRep + MeTA (Ours) | $75.89 \pm 0.4$ | $82.56 \pm 0.3$ | $90.52 \pm 0.2$ | $96.59 \pm 0.2$ | $85.67 \pm 0.3$ | $94.13 \pm 0.2$ |
| TGN + MeTA (Ours) | $\mathbf{83.47 \pm 0.2}$ | $\mathbf{90.85 \pm 0.2}$ | $\mathbf{92.96 \pm 0.1}$ | $\mathbf{98.17 \pm 0.1}$ | $\mathbf{90.82 \pm 0.2}$ | $\mathbf{98.26 \pm 0.1}$ |

For transductive edge prediction , we take the state-of-the-art approaches for representation learning on temporal graphs: CTDNE [21], JODIE [16], DyRep [28], TGAT [39], and TGN [25] as the baselines for comparison [39]. We conduct the experiments for 100 trials with random weight initialization. We implement MeTA with the TGN models DyRep and TGN. Table 2 reports the results. Our MeTA improves the test accuracy of DyRep by 3.9% on MOOC, 1.0% on Reddit, 1.3% on Wikipedia, and TGN by 3.0% on MOOC, 1.8% on Reddit, 2.0% on Wikipedia. As a result, our MeTA achieves substantial improvements for DyRep and TGN.

In the inductive setting, we keep the baselines which support inductive learning for comparison. We conduct the experiments for 100 trials with random weight initialization. The results are reported in Table 3. We implement our MeTA with DyRep and TGN to study whether MeTA can improve the performance of TGNs under the inductive setting. We observe that MeTA improves the test accuracy of DyRep by 4.1% on MOOC, 1.0% on Reddit, 2.6% on Wikipedia, and TGN by 3.4% on MOOC, 1.5% on Reddit, 2.5% on Wikipedia. As a result, our MeTA method enhances DyRep and TGN to outperform the baseline methods in the inductive task.

Given the TGN models DyRep and TGN, our MeTA achieves consistent and substantial improvements on all datasets, thanks to the adaptive data augmentation offered by our MeTA method. MeTA effectively augments the input features while preserving the essential semantic information, which reduces the noise on more informative parts of the input features and increase the diversity of data augmentation on less informative parts with adaptive data augmentation magnitudes. Overall, the results above indicate that our methods are effective in improving the effectiveness of the popular TGN models in both transductive and inductive settings.

## 4.2 Node Classification

The task of node classification on temporal graphs is to predict the time-varying labels of nodes [16]. Table 4 reports the test ROC AUC. We observe that MeTA improves ROC AUC of DyRep by 2.1% on MOOC, 2.3% on Reddit, 2.4% on Wikipedia, and TGN by 2.2% on MOOC, 2.0% on Reddit, 2.5% on Wikipedia. As a result, our MeTA method enhances DyRep and TGN to outperform the baseline methods in the temporal node classification. This validates the importance of adaptive data augmentation and our DA strategies on enriching the input data for temporal graph learning. Our MeTA adaptively augments the input features with appropriate magnitudes, and thus effectively enhance TGNs on node classification.

Table 4: ROC AUC (%) on the test set for temporal node classification. We conduct 100 trials with random weight initialization. Mean (%) and standard deviations are reported.

| Method | MOOC | Reddit | Wikipedia |
|---|---|---|---|
| CTDNE [21] | $67.54 \pm 0.7$ | $59.43 \pm 0.6$ | $75.89 \pm 0.5$ |
| JODIE [16] | $76.31 \pm 1.6$ | $61.83 \pm 2.7$ | $84.84 \pm 1.2$ |
| TGAT [39] | $74.25 \pm 0.9$ | $65.56 \pm 0.7$ | $83.69 \pm 0.7$ |
| DyRep [28] | $75.32 \pm 1.3$ | $62.91 \pm 2.4$ | $84.59 \pm 2.2$ |
| TGN [25] | $77.73 \pm 0.7$ | $67.06 \pm 0.9$ | $87.81 \pm 0.3$ |
| DyRep + MeTA (Ours) | $76.88 \pm 1.1$ | $64.36 \pm 2.0$ | $86.65 \pm 1.9$ |
| TGN + MeTA (Ours) | $\mathbf{79.41 \pm 0.8}$ | $\mathbf{68.37 \pm 0.9}$ | $\mathbf{90.03 \pm 0.3}$ |

## 4.3 Efficiency and Effectiveness with MeTA

As analyzed in Sec. 3.4 , MeTA enhance TGNs without increasing their time complexity. We evaluate the efficiency and effectiveness of TGN with MeTA using the Reddit dataset, which holds the largest number of nodes and second largest number of edges among the used datasets. We follow the experimental setting of the transductive edge prediction in Sec. 4.1.

The methods we evaluate include: TGN, our MeTA method of the level number $L = 1, 2$, and the refeed method to achieve adaptive DA introduced in Sec. 3.2. The number of memory updates (# Memory Updates) processed per epoch by different methods and the performance is reported in Table 5, where 'Time' is the training time until convergence using a Linux Server with an Intel(R) Xeon(R) E5-1650 v4 @ 3.60GHz CPU and a GeForce GTX 1080 Ti GPU. We notice that, compared with the refeed method, our MeTA significantly reduces the number of memory updates per epoch.

As a result, our MeTA method takes much less training time and memory updates than the refeed method, and exhibits similar efficiency to the original TGNs without DA, while the running time and test accuracy of refeed are unavailable (denoted as N.A.), since it takes too much time to converge. This agrees with the theoretical results in Sec. 3.4, since TGNs' time complexity is $\mathcal{O}(d \cdot \#\mathcal{E})$, which is the same as that of TGN with MeTA, while the time complexity of refeed is $\mathcal{O}(d(\#\mathcal{E})^2)$, much higher than our MeTA. Here, $\#\mathcal{E}$ is the number of edges, which is generally a large value in practice.

Table 5: Training time and test accuracy of TGN and TGN with DA on the Reddit dataset. The task is transductive edge prediction. We denote the refeed method for DA (see Sec. 3.2) as refeed. $L$ is the level number of our MeTA method. We report mean values over 100 trials with random weight initialization.

| Method | # Memory Updates | Time | Accuracy (%) |
|---|---|---|---|
| TGN [25] | $4.7 \times 10^5$ | 4625s | 92.56 |
| TGN + refeed | $1.1 \times 10^{11}$ | N.A. | N.A. |
| TGN + MeTA ($L = 1$) | $4.7 \times 10^5$ | **4513s** | 93.07 |
| TGN + MeTA ($L = 2$) | $9.4 \times 10^5$ | 7241s | **94.19** |

In terms of effectiveness, both MeTA ($L = 1$) and MeTA ($L = 2$) lead to substantial improvements over TGNs. Our DA techniques without MeTA enhances TGN to perform better than the original TGN without DA, since it enriches the input data with diverse DA strategies. Furthermore, our MeTA ($L = 2$) leads to more significant improvements than only the DA techniques with MeTA of $L = 1$, which validates the importance of adaptive DA achieved by MeTA. Our MeTA applies higher DA magnitudes to less important features, so as to effectively augment the input features and preserve the essential semantic information. Our DA strategies enhance the generalization of TGNs by applying diverse DA strategies to enrich the input data, while our META improves TGNs further by effectively augmenting the input features and preserving the essential semantic information with adaptive DA magnitudes.

### 4.4 Ablation Study

We investigate the contributions of our data augmentation strategies. We apply different DA strategies sequentially with our MeTA on the DyRep model. The results are presented in Table. 6. Our data augmentation strategies improve the effectiveness of DyRep by modeling the realistic noise to enrich the training data. Among different DA strategies, adding edges with time perturbation leads the largest

Table 6: Effects of our different DA strategies on the inductive edge prediction of MOOC.

| Technique | Accuracy (%) | $\Delta$ | Cumu $\Delta$ |
|---|---|---|---|
| DyRep | 72.92 | 0 | 0 |
| + Remove Edges | 73.64 | +0.72 | +0.72 |
| + Perturb Time | 74.68 | +1.04 | +1.76 |
| + Add Edges | **75.89** | **+1.21** | **+2.97** |

benefits, since it augments both the temporal and topology features. Perturbing time gives higher improvements than removing edges, because augmenting the temporal information meets the characteristics of the temporal graph data better.

Next, we evaluate the contributions of two message passing mechanisms in our MeTA module. We apply the cross-level propagation and memory transition mechanisms in our MeTA sequentially on the DyRep model. The results are presented in Table. 7. The row (+DA Strategies) means that we apply our DA strategies without the adaptive

Table 7: Effects of our different message passing mechanisms on the transductive edge prediction of MOOC.

| Mechanism | Accuracy (%) | $\Delta$ | Cumu $\Delta$ |
|---|---|---|---|
| DyRep | 73.36 | 0 | 0 |
| + DA Strategies | 73.67 | +0.31 | +0.31 |
| + Cross-level Propagation | 74.76 | +1.09 | +1.40 |
| + Memory Transition | 76.21 | **+1.45** | **+2.85** |

DA magnitudes provided by MeTA. Our message passing mechanisms improve the effectiveness of DyRep by providing the augmented data adaptive to topology and time. In our two mechanisms, memory transitions provide more improvements than the cross-level propagation. Recall that memory transitions adapt the DA magnitudes with respect to the time distances. This result shows that the time distances matter more than the topology distances on the adaptive DA for the temporal data, possibly because the time information meets the characteristics of the temporal graph data better.

Finally, we analyze the sensitivity of MeTA to the hyper-parameter: $p_1, p_2$ to control the DA magnitudes. The result is visualized in Fig. 3. We alter $p_1$ among $\{0, 0.1, 0.2, 0.3, 0.4, 0.5\}$ and $p_2$ among $\{0.5, 0.6, 0.7, 0.8, 0.9, 1\}$. The performance of TGN with MeTA is relatively smooth when parameters are within certain ranges. However, extremely large values of $p_1$ and small $p_2$ results in poor performances. Too large DA magnitudes on level 1 with large $p_1$ increase the risks of breaking the essential semantic information, while too small magnitudes on level 2 with small $p_2$ cannot augment the input features sufficiently. Moreover, only a poorly set hyper-parameter does not lead to significant performance degradation, which demonstrates that our MeTA framework is able to provide appropriate DA magnitudes for different parts of input features. We provide additional experiments about the hyper-parameter $T$ to control the memory transitions in Appendix.

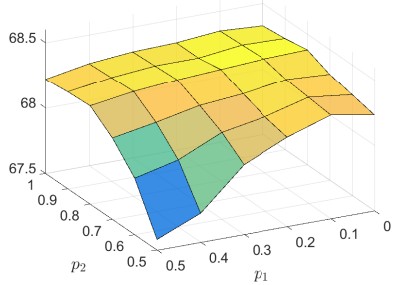

Figure 3: The ROC AUC (% in z-axis) of TGN with MeTA on node classification of Reddit with different hyperparameters $p_1$ and $p_2$.

## 5 Conclusion

In this paper, we study the problem of exploring data augmentation to strengthen the TGN models. We propose a novel methodology named MeTA, which provides adaptively augmented inputs for temporal graph learning. MeTA is a generic framework that can use any DA strategies to enhance popular TGN models. In our work, considering the characteristics of temporal graph data, we propose three DA strategies to augment the temporal graphs by modifying both temporal and topology features. Our experimental results show that MeTA, with our DA strategies, yields significant gains for both edge prediction and node classification in an efficient manner. One limitation of our methods is that our DA strategies cannot model all realistic noise on temporal graphs in practice. Therefore, future work could explore other advanced DA strategies to enhance our approach.

## Acknowledgements

This paper is supported by NUS ODPRT Grant R252-000-A81-133 and Singapore Ministry of Education Academic Research Fund Tier 3 under MOEs official grant number MOE2017-T3-1-007.

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
