# A Appendix

## A.1 Additional Experiments

We analyze the sensitivity of MeTA to the hyper-parameter: $T$ to control the period of memory transitions and $p_1, p_2$ to control the DA magnitudes. The results on $T$ are presented in Table 8. When $T$ is larger, more recent edges are assigned small DA magnitudes, so that the essential semantic information is preserved. Too large $T$ places too many edges on the first level, which may not effectively augment the input features. Note that in all of our experiments, we set $T = 10$ by default, which generally achieves satisfactory performance.

Table 8: Effects of different transition periods of MeTA on node classification.

| Transition Period | MOOC | Reddit | Wikipedia |
|---|---|---|---|
| $T = 2$ | 78.62 | 67.81 | 89.30 |
| $T = 5$ | 79.25 | 68.22 | 89.78 |
| $T = 10$ | **79.41** | 68.37 | **90.03** |
| $T = 20$ | 79.33 | **68.39** | 89.97 |

## A.2 Proof for Analysis and Discussion

We analyze how our MeTA and DA techniques work for temporal graph learning. Recall that if an edge is more distant to a target node on either topology or time, it is less important or informative to predict the target node's activities and thus we are expected to apply a higher DA magnitude to it. Next, we provide therotical analysis to show that our methods meet this expectation.

**Proposition 4** (Effects of topology). *If edge $\mathbf{e}_{km}^{(l)}(\tau)$ contribute to the node state $\mathbf{s}_i^{(1)}(t)$, and the distances from nodes $k$ and $m$ to $i$ are equal to or larger than $h$ hops in graph $\mathcal{G}$, we have the level of edge $\mathbf{e}_{km}^{(l)}(\tau)$: $l \geq \min(h + 1, L)$ and the time $\tau < t$, where $L$ is the number of levels.*

*Proof.* Denote $l_1 = l$ and $\tau_1 = \tau$. Since edge $\mathbf{e}_{km}^{(l)}(\tau)$ conributes to the node state $\mathbf{s}^i(t)$, from Alg. 1, there exists a temporal path $[\mathbf{e}_{km}^{(l_1)}(\tau_1), \ldots, \mathbf{e}_{oi}^{(l_N)}(\tau_N)]$ connecting $\mathbf{e}_{km}^{(l_1)}(\tau)$ and node $i$ to propagate the feature of $\mathbf{e}_{km}^{(l_1)}(\tau)$ to node $i$ at time $t$, where $N$ is the length of this path. Because of Eq. (2) and Eq. (3), $l_x > l_{x+1}$ and $\tau_x < \tau_{x+1} < t$ holds. Because the distances from nodes $k$ and $m$ to $i$ are equal or larger than $h$ hops in graph $\mathcal{G}$ and our DA techinique do not connect any node pairs that are not connected in $\mathcal{G}$, $N \geq h + 1$ holds. Because $l_N \geq 1, N \geq h + 1, l_x > l_{x+1}$ holds, there is $l = l_1 \geq h + 1$. Because $\tau_x < \tau_{x+1} < t$ holds, there is $\tau = \tau_1 < t$ □

Proposition 1 implies that the edges more distant to the target node on the topology must be on levels of higher $l$, and thus of higher DA magnitudes. In addition to the topology, we provide the following proposition to illustrate the effects of the temporal distances.

**Proposition 5** (Effects of time). *If edges $\mathbf{e}_{ij}^{(l_1)}(\tau_1)$ and $\mathbf{e}_{ik}^{(l_2)}(\tau_2)$ connects node $i$ and directly contribute to the node state $\mathbf{s}_i(t)$ as Eq. (2), we have $l_1 \geq l_2, \tau_1 < \tau_2 < t$ when $\tau_1 < \tau_2$ holds, and $l_1 \leq l_2, \tau_2 < \tau_1 < t$ holds with $\tau_1 > \tau_2$.*

*Proof.* Based on Alg. 1, because edges $\mathbf{e}_{ij}^{(l_1)}(\tau_1)$ and $\mathbf{e}_{ik}^{(l_2)}(\tau_2)$ directly contribute to the node state $\mathbf{s}^i(t)$ as Eq. (2), $l_1 = l_2$ holds if no memory transition of node $i$ happens between $\tau_1$ (inclusive) and $\tau_2$ (inclusive). If memory transition of node $i$ happens between $\tau_1$ (inclusive) and $\tau_2$ (inclusive), there is $l_1 > l_2$ with $\tau_1 < \tau_2$ and $l_1 < l_2$ with $\tau_1 > \tau_2$. Because edges $\mathbf{e}_{ij}^{(l_1)}(\tau_1)$ and $\mathbf{e}_{ik}^{(l_2)}(\tau_2)$ directly contribute to the node state $\mathbf{s}^i(t)$ as Eq. (2), $\tau_1 < t, \tau_2 < t$ holds. Overall, there is $l_1 \geq l_2, \tau_1 < \tau_2 < t$ when $\tau_1 < \tau_2$ holds, and $l_1 \leq l_2, \tau_2 < \tau_1 < t$ holds with $\tau_1 > \tau_2$. □

Proposition 2 shows that the earlier edges tend to be on levels of higher $l$, and thus of higher DA magnitudes. Then, we analyze how our DA strategies retain the original temporal graph characteristics to model the realistic noise. Many social and natural activities follow the poisson process [19, 13]. Denote the time range of the training data as $[0, T_{\max}]$. If nodes' interactions follow the poisson process, we have the following lemma for the distribution of the interactions' happening time [6]:

**Lemma 1.** *If interactions follow a homogenous poisson process, the happening time of an interaction follows the uniform distribution on the time range $[0, T_{\max}]$.*

Our data augmentation includes the perturbations on the edge time. We provide the following theorem to show that our DA techniques do not change the distribution of edge time.

**Theorem 2.** *If the edges before data augmentation follow a homogeneous Poisson process, our data augmentation does not change the distribution of any edge's occurrence time.*

*Proof.* From Lemma 1, we have the probability density function of the happening time of an edge as: $f(t) = \frac{1}{T_{\max}}, \forall t \in [0, T_{\max}]$. Denote the probability density function of the happening time of an event after the data augmentation as $f_{\mathrm{DA}}(t)$, the probability density function of the noise on edge time applied by our DA as $f_{\mathrm{noise}}(t)$, and $dt$ as an infinitely small value. Given an observed edge, we have the probability of the edge happening between $t$ and $t + dt$ after our data augmentation as

$$f_{\mathrm{DA}}(t)dt = \int_0^{T_{\max}} f(\tau) \cdot f_{\mathrm{noise}}(t - \tau)dtd\tau + \int_{-\infty}^0 f(t) \cdot f_{\mathrm{noise}}(\tau - t)dtd\tau + \tag{6}$$

$$\int_{T_{\max}}^{\infty} f(t) \cdot f_{\mathrm{noise}}(\tau - t)dtd\tau \tag{7}$$

$$= \int_0^{T_{\max}} \frac{1}{T_{\max}} f_{\mathrm{noise}}(t - \tau)dtd\tau + \int_{-\infty}^0 \frac{1}{T_{\max}} f_{\mathrm{noise}}(\tau - t)dtd\tau + \tag{8}$$

$$\int_{T_{\max}}^{\infty} \frac{1}{T_{\max}} f_{\mathrm{noise}}(\tau - t)dtd\tau \tag{9}$$

$$= \int_0^{T_{\max}} \frac{1}{T_{\max}} f_{\mathrm{noise}}(\tau - t)dtd\tau + \int_{-\infty}^0 \frac{1}{T_{\max}} f_{\mathrm{noise}}(\tau - t)dtd\tau + \tag{10}$$

$$\int_{T_{\max}}^{\infty} \frac{1}{T_{\max}} f_{\mathrm{noise}}(\tau - t)dtd\tau \tag{11}$$

$$= \frac{1}{T_{\max}} dt \left[ \int_0^{T_{\max}} f_{\mathrm{noise}}(\tau - t)d\tau + \int_{-\infty}^0 f_{\mathrm{noise}}(\tau - t)d\tau + \int_{T_{\max}}^{\infty} f_{\mathrm{noise}}(\tau - t)d\tau \right] \tag{12}$$

$$= \frac{1}{T_{\max}} dt \int_{-\infty}^{\infty} f_{\mathrm{noise}}(\tau - t) = \frac{1}{T_{\max}} dt \tag{13}$$

Therefore, we have:

$$f_{\mathrm{DA}}(t) = \frac{1}{T_{\max}}, \forall t \in [0, T_{\max}], \tag{14}$$

which is still a uniform distribution on the time range $[0, T_{\max}]$. $\square$

This theorem guarantees that our DA techniques do not break the original edge time distribution. Otherwise, if the original distribution is broken, e.g., all the edge time decrease to 0, the augmented edges cannot reflect the practical condition and may degrade the models' generalization.

In addition to the time distribution, we provide the following proposition to show the expected edge number does not change after our DA. Denote the edge set of the original graph $\mathcal{G}$ as $\mathcal{E}$ and that of the augmented graph $\mathcal{G}^{(l)}$ as $\mathcal{E}^{(l)}$.

**Proposition 6.** *After our data augmentation, the expected edge number is the same as that before data augmentation for any data augmentation magnitude, i.e., $\mathbb{E}[\#\mathcal{E}^{(l)}] = \#\mathcal{E}, \forall p_l \in [0, 1]$ holds.*

*Proof.* Denote the original graph as $\mathcal{G}$, and the corresponding edge set as $\mathcal{E}$. Our time perturbation DA technique does not change the number of edges. After removing edges, the augmented graph $\hat{\mathcal{G}}$ has the expected number of edges as

$$\mathbb{E}[\#\hat{\mathcal{E}}_1] = \sum_{i=1}^{\#\mathcal{E}} 1 - p = (1 - p)\#\mathcal{E} \tag{15}$$

After adding edges with the budget $p\#\mathcal{E}$, the number of edges in the augmented graph is:

$$\mathbb{E}[\#\hat{\mathcal{E}}] = \mathbb{E}[\#\hat{\mathcal{E}}_1 + p\#\mathcal{E}] = (1 - p)\#\mathcal{E} + p\#\mathcal{E} = \#\mathcal{E} \tag{16}$$

$\square$

This proposition shows that our data augmentation does not change the density of interactions in expectation, which meets the practical condition and does not induce extra computation load.

### A.3 More Details about Experiments for Reproducibility

In this section, we describe more detailed settings about the experiments to help in reproducibility.

#### A.3.1 Datasets and Software Versions

We download the MOOC dataset from the website[1], the Reddit dataset from the website[2], and the Wikipedia dataset from the website[3].

The Reddit dataset consists of one month of posts made by users on subreddits[4]. The collectors select the 1,000 most active subreddits as items and the 10,000 most active users. This results in 672,447 interactions. The collectors convert the text of each post into a feature vector representing their LIWC categories. The Reddit dataset holds ground-truth labels of banned users from Reddit[5]. This gives 366 true labels among 672,447 interactions (= 0.05%). The Wikipedia dataset consists of one month of edits made by edits on Wikipedia pages[6]. The collectors select the 1,000 most edited pages as items and editors who made at least 5 edits as users (a total of 8,227 users). This generates 157,474 interactions. Similar to the Reddit dataset, the collectors convert the edit text into an LIWC-feature vector. The MOOC dataset consists of actions, e.g., viewing a video, submitting an answer, etc., done by students on a MOOC online course[7]. This dataset consists of 7,047 users interacting with 98 items (videos, answers, etc.) resulting in over 411,749 interactions. There are 4,066 drop-out events (= 0.98%). The MOOC dataset was originally collected for the KDD 2015 challenge[8].

For the transductive setting, we examine embeddings of the nodes that have been observed in training, via the future edge prediction task and the node classification. In the inductive setting, we examine the inductive learning capability using the inferred representations of unseen nodes, by predicting the future links between unseen nodes and classify them based on their inferred embedding dynamically. For the inductive experiments, as suggested by [25] and [39], we randomly sample 10% of nodes, mask them during training and treat them as unseen nodes by only considering their interactions in validation and testing period. As such, an appropriate number of future edges among the unseen nodes will show up during validation and testing.

Regarding software versions, we install CUDA 10.0 and cuDNN 7.0. TensorFlow 1.12.0 and PyTorch 1.0.0 with Python 3.6.0 are used. Note that all the experiments are running a Linux Server with the Intel(R) Xeon(R) E5-1650 v4 @ 3.60GHz CPU, and the GeForce GTX 1080 Ti GPU.

#### A.3.2 Settings of Baseline

**CTDNE.** For the temporal network embedding model CTDNE, the walk length for the temporal random walk is selected among $\{60, 80, 100\}$, where setting walk length to 80 gives slightly better validation outcome. The original paper considers several temporal edge selection (sampling) methods (uniform, linear and exponential) and finds uniform sampling with best performances [21]. Since our setting is similar to theirs, we adopt the uniform sampling approach.

**TGAT.** As suggested by the authors, we fix the node embedding dimension and the time encoding dimension to be the original feature dimension for simplicity, and then select the number of TGAT layers from $\{1, 2, 3\}$, the number of attention heads from $\{1, 2, 3, 4, 5\}$, according to the link prediction AP score in the validation dataset. Although TGAT does not put restriction on the neighborhood size during aggregations, to speed up training, especially when using the multi-hop aggregations, we use neighborhood dropout (selected among $p = \{0.1, 0.3, 0.5\}$) with the uniform sampling. During training, we use 0.0001 as the learning rate for all datasets, with Glorot initialization and the Adam

---

[1]http://snap.stanford.edu/jodie/mooc.csv

[2]http://snap.stanford.edu/jodie/reddit.csv

[3]http://snap.stanford.edu/jodie/wikipedia.csv

[4]http://files.pushshift.io/reddit/

[5]https://www.reddit.com/

[6]https://meta.wikimedia.org/wiki/Data_dumps

[7]https://www.mooc.org/

[8]https://biendata.com/competition/kddcup2015/data/

SGD optimizer. Based on the validation results, using two TGAT layers and two attention heads with dropout rate of 0.1 gives the best performance. For inference, we inductively compute the embeddings for both the unseen and observed nodes at each time point that the graph evolves, or when the node labels are updated. We then use these embeddings as features for the future link prediction and dynamic node classifications with multilayer perceptron.

**JODIE, DyRep, and TGN.** When implementing JODIE, DyRep, and TGN as the benchmark, we use the Adam optimizer with a learning rate of 0.0001, a batch size of 200 for both training, validation and testing, and early stopping with a patience of 5. We sample an equal amount of negatives to the positive interactions, and use average precision as reference metric. Additional hyper-parameters used for both future edge prediction and dynamic node classification are as following: memory dimension as 172, node embedding dimension 100, time embedding dimension 100, number of attention heads as 2, dropout ratio as 0.1. For all the graph embedding modules, as suggested by [25], we use neighbors sampling [10] (i.e. only aggregate from $k$ neighbors) since it improves the efficiency of the model without decreasing accuracy. As suggested by [25], the sampled edges are the $k$ most recent ones, rather than the traditional approach of sampling them uniformly, since we found it to perform much better on the validation set. For JODIE we simply use the time embedding module, while for DyRep we augment the messages with the result of temporal graph attention performed on the destination's neighborhood. For both we use a vanilla RNN as the memory updater module.

We refer to the following websites when implementing the above mentioned models:

1. **CTDNE**: https://github.com/LogicJake/CTDNE
2. **TGAT**: https://github.com/StatsDLMathsRecomSys/Inductive-representation-learning-on-temporal-graphs
3. **JODIE**: https://github.com/srijankr/jodie
4. **DyRep**: https://openreview.net/forum?id=HyePrhR5KX
5. **TGN**: https://github.com/twitter-research/tgn

## A.4 Broader impact

This article mainly discusses data augmentation on temporal graphs. One of the most popular scenarios in this field is the social network. Our methods support adaptively augmenting the temporal graph data. The proposed data augmentation method is simple, fast, and efficient. It is possible that the proposed algorithms provide an effective solution on how to augment the temporal graphs for better effectiveness on temporal graph models. Temporal graph learning is generally being applied to more and more tasks and applications, since many practical scenarios can be modeled as evolving topology data. Some of the examples include recommendation systems, financial investments, and transportation analysis, etc. However, the study of risks of applying temporal graph learning methods, such as adversarial attacks, privacy protection, ethics and biases are still at an early stage. In practice, we should be warned about such risks and devise testing and monitoring framework carefully to avoid undesirable outcomes.