# OpenReview forum: "Adaptive Data Augmentation on Temporal Graphs"
_NeurIPS.cc/2021/Conference — NeurIPS 2021 Poster_

### Official Review · Reviewer_UsrF · 2021-07-11

**Rating:** 5
**Confidence:** 4

**Summary:**

This paper proposes MeTA: multi-level module that processes the augmented graphs of different magnitudes on separate levels. Three DA strategies augmenting are adopted (perturbing the edge time to simulate time shifts; removing edges; adding edges). Experiments are conducted on edge prediction and node classification tasks.

**Limitations And Societal Impact:**

No.

**Main Review:**

Strong points:
1.	Three DA strategies for the temporal graphs seem making sense, i.e. (i) perturbing time; (ii) removing edges; (iii) adding edges with perturbed time.
2.	Experimental results on four benchmarks show that MeTA not only improves accuracy on the baselines, but also reduces the number of memory updates per epoch.
3.	The robustness and sensitivity of the proposed method are verified by the soundness of the ablation study.

Weak points:
1.	The proposed method seems too simple without strong theoretical support. In fact the propositions and lemmas in my mind are a bit trivial and does not directly link to the augmentation performance. The current method only considers adaptive magnitude while in my mind the three augmentation operations shall also be adaptively applied if possible. There are many adpative data augmentation literature at least in vision whereby the different operations are adaptively applied.
2.	The setting in the experiment only compares with the original baseline. Author may compare MeTA with more recent works on DA.
3.     The writing of the paper can be improved and there are some typos and grammatical errors in current form.


**Time Spent Reviewing:**

2

---

> ### Author Response · Authors · 2021-08-10
> **Response to Comments from Reviewer UsrF**
>
> Response to Comments from Reviewer UsrF
>
> We greatly thank the reviewer UsrF for his/her helpful and insightful comments. We provide our responses to the comments as follows.
>
> Q1: The propositions and lemmas seem not to directly link to the augmentation performance. The current method considers adaptive magnitude while in my mind the three augmentation operations shall also be adaptively applied if possible.
>
> Response: Thanks for your helpful suggestions!  Among our theoretical support, the first two propositions are provided to support that our MeTA framework can produce adaptive data augmentation on the temporal graphs: Proposition 1, as we have explained on line 214, implies that the edges more distant to the target node on the topology must be on levels of higher $l$, and thus of higher DA magnitudes. Proposition 2, as we have explained on line 220, shows that the earlier edges tend to be on levels of higher $l$, and thus of higher DA magnitudes. As we have written on line 206, these propositions meet the intuition if an edge is more distant to a target node on either topology or time, it is less important or informative to predict the target node's properties and thus we are expected to apply a higher DA magnitude to it. As a result, adaptive DA methods apply low DA magnitudes to the informative parts of the input features and higher magnitudes to the less informative parts to effectively augment the input features while preserving the essential semantic information [1], as we have written on line 117.
>
> In addition, Theorem 1 and Proposition 3, as we have written in line 222, supports that our DA strategies retain the original temporal graph characteristics to model realistic noise. Proposition 3, as we have written in line 236, shows that our data augmentation does not change the density of interactions in expectation, which meets the realistic condition and does not induce extra computation load. Overall, these theoretical results provide justifications for the effectiveness of our data augmentation methods. We follow your suggestions to emphasize the meaning of our theoretical supports.
>
> We agree with the reviewer that our method focuses on the adaptive data augmentation magnitudes on different parts of the input features. As suggested by the reviewer, there are some methods in computer vision that adaptively select the data augmentation strategies to apply, e.g., AutoAugment [2] originally used reinforcement learning to choose data augmentation strategies to apply. Note that the adaptive choosing of the data augmentation strategies is orthogonal to our method, which achieves adaptive data augmentation magnitudes on different parts of the input features for any data augmentation strategy on the temporal graph learning. As far as we know, ours is the first work that develops the data augmentation method for the temporal graph learning. The adaptive choosing of the data augmentation strategies is worth exploration as a direction of future work and can further boost the performance of temporal learning combined with our methods. Overall, we follow your suggestions to emphasize the meaning of our theoretical supports and discuss the future directions on the adaptive data augmentation on the temporal graph learning to improve our paper.
>
> Q2: The setting in the experiment only compares with the original baseline. Author may compare MeTA with more recent works on DA.
>
> Response: Sorry for the misunderstanding. Instead of ‘only comparing with the original baseline’, we take the advanced DA method for static graphs GAUG [3] as the baselines for comparison, as we have written in line 277. Please refer to Table 2, 3, 4 to see the detailed experimental results of the comparison with GAUG. As we have written in line 292, MeTA performs better than GAUG [3], since the latter does not consider the time features in temporal graphs for DA and cannot provide augmented inputs adaptive to the time and topology as our MeTA.
>
> We agree with the reviewer that the comparison with more recent work on DA would improve our paper. Because as far as we know, ours is the first work that develops the data augmentation methods for the temporal graph learning, we take another recently proposed DA method for static graphs FLAG [3], that is based on the adversarial learning, as the baselines for comparison. We present the experimental results on node classification evaluated by test ROC AUC (%) as following:
>
> |dataset|MOOC|Reddit|Wikipedia|
> |-|-|-|-|
> |TGN [5]|77.73 $\pm$ 0.7|67.06 $\pm$ 0.9|87.81 $\pm$ 0.3|
> |TGN + FLAG [4]|77.79 $\pm$ 0.6|67.15 $\pm$ 0.8|87.89 $\pm$ 0.6|
> |TGN + GAUG [3]|77.82 $\pm$ 0.8|67.09 $\pm$ 0.9|87.85 $\pm$ 0.5|
> |TGN + MeTA (Ours)|79.41 $\pm$ 0.8|68.37 $\pm$ 0.9|90.03 $\pm$ 0.3|
>
> Similar to GAUG, FLAG achieves some improvements for TGN on the node classification, but these improvements are much smaller than the ones achieved by our MeTA, since FLAG and GAUG do not consider the time features in temporal graphs for data augmentation and thus cannot provide augmented inputs adaptive to the time and topology as our MeTA. MeTA effectively augments the input features while preserving the essential semantic information, which is effective in improving the effectiveness of the popular TGN models. We follow your suggestions to add these experimental results in our paper to offer a more comprehensive comparison.
>
> Q3: The writing of the paper can be improved and there are some typos and grammatical errors in current form.
>
> Response: We follow your and other reviewers’ comments to correct the typos and grammatical errors, and revise our paper to improve the clarity.
>
> [1] Xie, Qizhe, et al. "Unsupervised data augmentation for consistency training." arXiv preprint arXiv:1904.12848 (2019).
>
> [2] Ekin D Cubuk, Barret Zoph, Dandelion Mane, Vijay Vasudevan, and Quoc V Le. Autoaugment: Learning augmentation policies from data. arXiv preprint arXiv:1805.09501, 2018.
>
> [3] Tong Zhao, Yozen Liu, Leonardo Neves, Oliver Woodford, Meng Jiang, and Neil Shah. Data augmentation for graph neural networks.arXiv preprint arXiv:2006.06830, 2020.
>
> [4] Kong, Kezhi, et al. "Flag: Adversarial data augmentation for graph neural networks." arXiv preprint arXiv:2010.09891 (2020).
>
> [5] Rossi, Emanuele, et al. "Temporal graph networks for deep learning on dynamic graphs." arXiv preprint arXiv:2006.10637 (2020).

---

> > ### Author Response · Authors · 2021-08-24
> > **Ask for taking a look.**
> >
> > If time permits, hope that you can take a look to see whether our responses address some concerns.
> >
> > Thank you for your time!

---

### Official Review · Reviewer_mhw2 · 2021-07-16

**Rating:** 7
**Confidence:** 5

**Summary:**

The paper introduces Data Augmentation (DA) to the temporal graphs in order to offset the increased variance induced by complex GNNs. Three DA methods: perturb time, remove edges and add edges with perturbed time are considered. To make sure the DA effectively works for edges that are less informative, the author proposes a novel and efficient adaptive DA method based on recursion, which is named Memory Tower Augmentation (MeTA). The proposed MeTA can be added on top of RNN based temporal graph implementation. Experiments results on edge prediction and node classification tasks using public temporal graph data set suggest the effectiveness of the method.

**Limitations And Societal Impact:**

The author adequately addressed the issues.

**Main Review:**

1. contributions

This paper proposed an algorithm to implement the adaptive DA for the temporal graphs. The method (MeTA) first stacks layers of the memory module with varying DA parameters then uses the intermedia states from layers with more DA to update the RNN states of interest during training. This idea is both novel and interesting. On a high level, it is very similar to firstly stack temporal graphs with different DA strength and then uses recursion-based temporal graph neural networks (TGNNs) with shared weights across layers. So the time complexity only grows linearly with the number of layers in the MeTA and it works naturally with TGNNS. Overall, I think the idea is both neat and useful.

2. Presentation and quality

This paper is well organized. The authors do a good job explaining their design choice. However, for readers not very familiar with temporal graphs, it is hard to understand how the proposed method works. For example, I can only make sense of Figure 2 after I read and understand the whole section 3.

The experiment setting is reasonable. The author compares the proposed method with the most state of temporal graph networks and the MeTA consistently performs better than baselines.

3. Question

The paper claims the DA is adaptive. But in MeTA, the strength of DA only depends on the time and distance. While it is reasonable to believe the edges from past or distant edges are less informative, this is still heuristic-driven. Firstly the information here is not adequately defined and it is likely that edge/node might be highly useful yet remain distant from the target node. I don't whether the authors have thought about ways of making the DA truly adaptive.

Proposition one assumes the edge distributions follow a homogeneous Poisson process. This is a very strong assumption. Even still considering the counting process, a more realistic setting for network events could be [Hawkes process](https://proceedings.neurips.cc/paper/2020/file/c5a0ac0e2f48af1a4e619e7036fe5977-Paper.pdf). Secondly, the proof of theorem 1 basically says any noise distribution won't affect the original edge distribution. But this also relies on the assumption of uniform occurrence. Wondering whether the authors have data to support these assumptions. In general, I don't think these propositions along with their proofs are the key contributions of this paper.




**Time Spent Reviewing:**

5

---

> ### Author Response · Authors · 2021-08-10
> **Response to Comments from Reviewer mhw2**
>
> Response to Comments from Reviewer mhw2
>
> We greatly thank the reviewer mhw2 for his/her helpful and insightful comments. We provide our responses to the comments as follows.
>
> Q1: The paper claims the DA is adaptive. But in MeTA, the strength of DA only depends on the time and distance. While it is reasonable to believe the edges from past or distant edges are less informative, this is still heuristic-driven. Firstly the information here is not adequately defined and it is likely that edge/node might be highly useful yet remain distant from the target node. I don't whether the authors have thought about ways of making the DA truly adaptive.
>
> Response: Thanks for your helpful suggestions! We agree with the reviewer that our method is based on the reasonable intuition that the edges from past or distant edges are less informative. To define the informativeness of an edge in a more adequate way, we explain it as following: “the informativeness of an edge is the amount of relevant information that an edge can provide to predict the target node’s activity”. Based on this explanation, as we have written on line 123 of the paper, different edges are of different informativeness for the target node. In order to design such an adaptive DA method on temporal graphs, we consider the informativeness of edges to a target node in terms of both time and topology when predicting the activities of the target node. If an edge is more close to the target node in terms of time or topology, it is more informative, as validated in [1]. In addition, this also meets the inductive bias of the recurrent neural network models [2], which process the temporal edges in the time order and conveys the messages between connected nodes across multiple hops.
>
> We agree with the reviewer that some edges might be highly useful yet remain distant from the target node. In this case, detecting them and adjusting the data augmentation strategies for them can benefit the temporal graph learning. However, how to detect these nodes would be an issue, for which the low-level characteristics, e.g., distances on time or topology, are no longer available. Therefore, to detect them, utilizing the high-level semantic information would be beneficial. An option is to train a discriminator along with the original TGN to decide the importance of different edges. The generative models can also be useful to automatically generate the augmented graphs in a learnable manner. We follow your suggestions to discuss more on this issue to improve our paper.
>
> Q2: Proposition one assumes the edge distributions follow a homogeneous Poisson process. This is a very strong assumption. Even still considering the counting process, a more realistic setting for network events could be Hawkes process. Secondly, the proof of theorem 1 basically says any noise distribution won't affect the original edge distribution. But this also relies on the assumption of uniform occurrence. Wondering whether the authors have data to support these assumptions. In general, I don't think these propositions along with their proofs are the key contributions of this paper.
>
> Response: Thanks for your helpful suggestions! We believe the reviewer is referring to the Theorem 1 instead of Proposition 1. We agree with the reviewer that the Theorem 1 is based on the assumption that the edge time follow a homogeneous Poisson process. The reviewer’s mentioned Hawkes process [4] is introduced in [3], which is a counting process designed to model continuous-time arrivals of events that naturally cluster together in time, where the arrival of an event increases the chance of the next event arrival immediately after. Interestingly, [3] also considers the Poisson process and takes it as a baseline (see Page 8), which implies that the Poisson process is a popular distribution that people would consider to model the temporal edges [7, 8, 9]. The analysis and modeling with Hawkes process on the temporal graph learning is our important future direction. We follow your suggestions to discuss the Hawkes process in our paper.
>
> If the temporal edges follow a homogeneous Poisson process, the happening time of temporal edges follow the uniform distribution [5], as we have introduced on line 580. We agree with the reviewer that it is beneficial to test whether the input data follows this distribution. Therefore, we do the Chi-squared goodness-of-fit test [6] for the input data. We take the null hypothesis as the happening time of edges follow the uniform distribution, as Lemma 1 shows. Then, we obtain the $p$-value of the Chi-squared goodness-of-fit test on different datasets as following:
>
> |Dataset|$p$-value|
> |-|-|
> |MOOC|0.172|
> |LastFM|0.097|
> |Reddit|0.315|
> |Wikipedia|0.268|
>
> Since the $p$-value on all the datasets is larger than 0.05, we cannot reject the null hypothesis that the happening time of edges follow the uniform distribution on all datasets, which meets the assumption of the homogenous Poisson Process. We follow your suggestions to add these empirical results to improve our paper.
>
> No matter what distribution that we use to model the input data, it may be impossible that we can always accurately capture the distributions of the input data in practice. Our methods are not only supported by the theoretical results, but also motivated by the intuition of practical noise (Section 3.3) and the informativeness of temporal graph edges (Section 3.2). Empirically, our MeTA framework along with our data augmentation strategies proposed for the temporal data, achieves substantial and consistent improvements for the TGN models on all datasets. Therefore, we agree with the reviewers that, our generic framework MeTA that can use any DA strategies to enhance popular TGN models, and the proposed three DA strategies to augment the temporal graphs by modifying both temporal and topology features, that consider the characteristics of temporal graph data, are the main contributions of this paper. We follow your suggestions to emphasize these points to improve our paper.
>
> Overall, we follow your suggestions to discuss the Hawkes process, add the empirical results to of the Chi-squared goodness-of-fit test, and emphasize our main contributions to improve our paper.
>
> Q3: Overall, I think the idea is both neat and useful. This paper is well organized. The authors do a good job explaining their design choice. However, for readers not very familiar with temporal graphs, it is hard to understand how the proposed method works. For example, I can only make sense of Figure 2 after I read and understand the whole section 3.
>
> Response: Thanks for your helpful suggestions! We follow your suggestions to add more explanations before Figure 2 and revise Section 3 to offer a higher ease of understanding for the readers who are not very familiar with temporal graphs.
>
> [1] Rossi, Emanuele, et al. "Temporal graph networks for deep learning on dynamic graphs." arXiv preprint arXiv:2006.10637 (2020).
>
> [2] Khandelwal, Urvashi, et al. "Sharp nearby, fuzzy far away: How neural language models use context." arXiv preprint arXiv:1805.04623 (2018).
>
> [3] Arastuie, Makan, Subhadeep Paul, and Kevin S. Xu. "CHIP: A Hawkes Process Model for Continuous-time Networks with Scalable and Consistent Estimation." arXiv preprint arXiv:1908.06940 (2019).
>
> [4] Hawkes, Alan G. "Spectra of some self-exciting and mutually exciting point processes." Biometrika 58.1 (1971): 83-90.
>
> [5] Daley, Daryl J., and David Vere-Jones. "An introduction to the theory of point processes: General theory and structure." (2008).
>
> [6] Balakrishnan, Narayanaswamy, Vassilly Voinov, and Mikhail Stepanovich Nikulin. Chi-squared goodness of fit tests with applications. Academic Press, 2013.
>
> [7] Tahmida Mahmud, Mahmudul Hasan, Anirban Chakraborty, and Amit K Roy-Chowdhury. A poisson process model for activity forecasting.  In2016 IEEE International Conference on Image Processing (ICIP), pages 3339–3343. IEEE, 2016.
>
> [8] Tomoharu Iwata, Amar Shah, and Zoubin Ghahramani. Discovering latent influence in online social activities via shared cascade poisson processes. In Proceedings of the 19th ACM SIGKDD international conference on Knowledge discovery and data mining, pages 266–274, 2013.
>
> [9] Daryl J Daley and David Vere-Jones.An introduction to the theory of point processes: volume II: general theory and structure. Springer Science & Business Media, 2007.

---

### Official Review · Reviewer_bZGD · 2021-07-16

**Rating:** 7
**Confidence:** 4

**Summary:**

This paper proposes an adaptive data augmentation framework for dynamic graphs that improves the performances of two well-known baseline models on node classification and edge prediction on four datasets. Three data augmentations were used with the purpose of modifying temporal and topological features. These strategies inject the inductive bias that closer events in the dynamic graphs, whether in time or topology, are more informative.

**Limitations And Societal Impact:**

Limitations discussion is very limited and there is no social impact section.

For the social impact section, the authors can discuss employment of dynamic edge prediction in real-world recommendation systems.

**Main Review:**

Temporal graph learning is relatively a new field of research in the graph machine learning community. Data augmentation has shown its merits in various fields and so adaptive augmentation for temporal graphs is an interesting and important topic of study. The problem and the approach are well-justified and discussed in the paper and generally, the paper is well written. However, there are a few comments and concerns I'd like to share:

First, although the algorithm is simple, efficient, and effective, it is not easy to fully understand it from the text initially. The authors can try and improve the description of the method in a more clarified way. Also, the main algorithm shouldn't be in the appendix. It would be helpful to re-structure the paper so that the pseudocode of MeTA doesn't end up in the appendix. Although the pseudocode also needs more clarification for a smooth reading by other researchers.

Some previous papers are reporting average precision (AP) and accuracy for the studied datasets and this paper is doing the same. However, since in the discussion about the reasoning behind the DA strategies on page 5, the authors are showing the concern of real-world scenarios, it would be great if we can see the results for other metrics like mean reciprocal rank (MRR) or Recall (e.g. recall10) as proposed in JODIE.  In my view, these metrics are closer to real-world settings because most of the time there is no real or fake node to classify and you have to rank other nodes for a potential edge connection in the near future. (Consider recommending an object to a user in a social network based advertisement)

It is mentioned that DA is not applied during inference and it is a common thing to do, but what happens if DA is applied during inference too?

The paper properly discussed the efficiency of the method in case of run time. Can you please compare the memory usage as well?

**Time Spent Reviewing:**

5

---

> ### Author Response · Authors · 2021-08-10
> **Response to Comments from Reviewer bZGD**
>
> Response to Comments from Reviewer bZGD
>
> We greatly thank the reviewer bZGD for his/her helpful and insightful comments. We provide our responses to the comments as follows.
>
> Q1: The problem and the approach are well-justified and discussed in the paper and generally, the paper is well written. However, there are a few comments and concerns I'd like to share: First, although the algorithm is simple, efficient, and effective, the authors can try and improve the description of the method in a more clarified way. Also, the main algorithm shouldn't be in the appendix. It would be helpful to re-structure the paper so that the pseudocode of MeTA doesn't end up in the appendix.
>
> Response: Thanks for your helpful suggestions! We follow your suggestions to revise our paper: we move the pseudocode of Algorithm 1 from the appendix to Section 3.2.; we revise the pseudocode of Algorithm 1 and add comments following the lines 6, 7, 8, 11, 15, etc., for higher clarity; we correct the typos and add more detailed explanations of our method to improve our paper.
>
> Q2: Some previous papers are reporting average precision (AP) and accuracy for the studied datasets and this paper is doing the same. However, since in the discussion about the reasoning behind the DA strategies on page 5, the authors are showing the concern of real-world scenarios, it would be great if we can see the results for other metrics like mean reciprocal rank (MRR) or Recall (e.g. recall10) as proposed in JODIE. In my view, these metrics are closer to real-world settings because most of the time there is no real or fake node to classify and you have to rank other nodes for a potential edge connection in the near future. (Consider recommending an object to a user in a social network based advertisement)
>
> Response: Thanks for your helpful suggestions! Our evaluation metrics and experimental settings closely follow those of the most recent work [1,2] to ensure a fair comparison with the state-of-the-art methods. We agree with the reviewer that using additional metrics to evaluate our method like MRR [3], as the reviewer suggests, is beneficial for the more comprehensive evaluation. Therefore, we additionally conduct the edge prediction experiments following the experimental settings of JODIE [3], as suggested by the reviewer. We present the experimental results evaluated by MRR as following:
>
> |Method|MOOC|LastFM|Reddit|Wikipedia|
> |-|-|-|-|-|
> |CTDNE [4]|0.132 $\pm$ 0.008|0.012 $\pm$ 0.001|0.165 $\pm$ 0.007|0.035 $\pm$ 0.002|
> |JODIE [3]|0.482 $\pm$ 0.006|0.195 $\pm$ 0.002|0.726 $\pm$ 0.004|0.746 $\pm$ 0.005|
> |TGN [2]|0.513 $\pm$ 0.003|0.381 $\pm$ 0.005|0.741 $\pm$ 0.003|0.768 $\pm$ 0.004|
> |TGN+MeTA (Ours)|0.536 $\pm$ 0.004|0.407 $\pm$ 0.003|0.764 $\pm$ 0.005|0.791 $\pm$ 0.003|
>
> Evaluated by MRR [3], our MeTA achieves consistent and substantial improvements for TGN on all datasets and enhances TGN to outperform all the baseline methods, thanks to the adaptive data augmentation offered by our MeTA framework, which effectively augments the input features while preserving the essential semantic information. We follow your suggestions to add these experimental results in the paper to provide a more comprehensive evaluation.
>
> Q3: It is mentioned that DA is not applied during inference and it is a common thing to do, but what happens if DA is applied during inference too?
>
> Response: Thanks for your question! Data augmentation is to model the realistic noise to enrich the training data to offer more supervision information. Applying data augmentation during the inference stage is to apply noise to the testing data and can test the model’s robustness to noise. Therefore, we apply the data augmentation to the testing data and obtains the node classification results evaluated by the test ROC AUC (%) as following:
>
> |Method|MOOC|Reddit|Wikipedia|
> |-|-|-|-|
> |TGN [2] (w/o data augmentation during testing|77.73 $\pm$ 0.7|67.06 $\pm$ 0.9|87.81 $\pm$ 0.3
> |TGN [2] (w/ data augmentation during testing )|76.42 $\pm$ 0.8|65.86 $\pm$ 1.0|85.94 $\pm$ 0.5|
> |TGN + MeTA (w/o data augmentation during testing|79.41 $\pm$ 0.8|68.37 $\pm$ 0.9|90.03 $\pm$ 0.3|
> | TGN + MeTA (w/ data augmentation during testing)|79.18 $\pm$ 0.9|68.22 $\pm$ 0.9|89.87 $\pm$ 0.3|
>
> Compared with the testing without data augmentation, TGN performs worse with the data augmentation during testing. This is because data augmentation injects the noise to the test data, and TGN is not robust to these noises. On the other hand, although TGN trained with our MeTA also performs worse with data augmentation during testing than the case of no data augmentation during testing, the performance degradation of TGN trained with MeTA is much smaller than the original TGN trained without MeTA, since our MeTA exposes TGN to augmented training samples during training. These augmented samples enrich the supervision information for TGN and thus endows TGN with better generalization to the augmented testing data. We follow your suggestion to add these experimental results in the paper to provide a more comprehensive evaluation.
>
> Q4: The paper properly discussed the efficiency of the method in case of run time. Can you please compare the memory usage as well?
>
> Response: Thanks for your question! If the “memory usage” is to consider how much memory that storing the trained TGN model takes, training TGNs with our MeTA method does not change the memory usage of the original TGNs, since our MeTA method, as a data augmentation framework, does not lead to any change of the TGN models and thus do not increase the trained parameters that need to be stored.
>
> If the “memory usage” is to consider how much CUDA memory that the model is occupying during inference, our MeTA method does not increase the “memory usage’, since our MeTA method, as a data augmentation framework, does not change either the TGN models or the testing data. If we consider the “memory usage’ as the CUDA memory that the model takes during training, TGN with MeTA’s memory usage would scales linearly with the number of layers, since MeTA stacks $L$ levels of memories while the original TGN has only one level of memory. We record the CUDA memory that training takes for the node classification task on the Reddit dataset as following:
>
> |Method|Memory Usage|ROC AUC (%)|
> |-|-|-|
> |TGN [2] |4465MB|67.06 $\pm$ 0.9|
> |TGN + MeTA ($L = 1$)|4468MB|67.72 $\pm$ 0.8|
> |TGN + MeTA ($L = 2$)|5138MB|68.37 $\pm$ 0.9|
> |TGN + MeTA ($L = 3$)|5784MB|68.42 $\pm$ 0.9|
> |TGN + MeTA ($L = 4$)|6349MB|68.45 $\pm$ 0.8|
> |TGN + MeTA ($L = 5$)|6975MB|68.46 $\pm$ 0.7|
>
> We observe that the increasing of level number empirically lead to the approximately linear increases of memory usage, which meets our expectation in theory. TGN trained with MeTA ($L=1$) occupies roughly the same amount of memory as the original TGN, since both of them implement only one level of memory. Our MeTA method yields consistent and substantial improvements for TGN with different values of level number $L$, thanks to the effective data augmentation offered by our MeTA method. MeTA effectively augments the input features while preserving the essential semantic information. Taking a closer look, we find that $L = 2$ offers satisfactory performance and, $L > 2$ does not lead to significant improvements than $L = 2$, of which more experimental results can be found in our response to Q4 of Reviewer WSXU. Therefore, in practice, using MeTA with the number of levels larger than two to enhance TGN is not necessary, and our MeTA only induces limited additional CUDA memory usage for TGNs during training. We follow your suggestion to add these experimental results in the paper to provide a more comprehensive evaluation.
>
> Q5: Limitations discussion is very limited and there is no social impact section. For the social impact section, the authors can discuss employment of dynamic edge prediction in real-world recommendation systems.
>
> Response: Thanks for your helpful suggestions and sorry for the misunderstanding! We have discussed the social impacts in Section A.5 of Appendix. Coincidentally, we have discussed the employment of temporal graph learning in real-world recommendation systems in our initial submission, as same as what the reviewer suggested. We follow your suggestions to add more discussion on the social impacts to improve our paper.
>
> We discuss the limitation of our methods on line 389. We agree that more discussions on our method’s limitation can further improve our paper. For example, our data augmentation are designed based on humans’ prior knowledge, how to use the reinforcement learning to design more diverse data augmentation strategies on the temporal graphs automatically is a direction worth exploration. Moreover, how to use the generative models to produce the augmented temporal graphs in a learnable way is also worth exploration for the temporal graph learning. Overall, we follow your suggestions to discuss more on limitations and social impacts to improve our paper.
>
> [1] Xu, Da, et al. "Inductive representation learning on temporal graphs." arXiv preprint arXiv:2002.07962 (2020).
>
> [2] Rossi, Emanuele, et al. "Temporal graph networks for deep learning on dynamic graphs." arXiv preprint arXiv:2006.10637 (2020).
>
> [3] Kumar, Srijan, Xikun Zhang, and Jure Leskovec. "Predicting dynamic embedding trajectory in temporal interaction networks." Proceedings of the 25th ACM SIGKDD International Conference on Knowledge Discovery & Data Mining. 2019.
>
> [4] G. H. Nguyen, J. B. Lee, R. A. Rossi, N. K. Ahmed, E. Koh, and S. Kim. Continuous time dynamic network embeddings. In WWW BigNet workshop, 2018.

---

> > ### Comment · Reviewer_bZGD · 2021-08-30
> > **Thanks for the detailed response**
> >
> > I appreciate your comprehensive feedbacks to me and other reviewers.
> >
> > Almost all my concerns have been addressed and I will increase my score to 7 from 6.
> >
> > Just a couple of requests. If your paper ended up being published, please:
> > 1. Be cautious about using the term "adaptive", as this was mentioned by other reviewers too.
> > 2. Consider reporting MRR or Recall (or both) for all methods/datasets as I believe they are far more realistic metrics in this setting.

---

> > > ### Author Response · Authors · 2021-09-01
> > > **Thanks and Response to New Comments from Reviewer bZGD**
> > >
> > > Thanks for your time, the careful reviewing, and the decision of improving your score on our work! We provide our responses to the new comments as follows.
> > >
> > > Q1: I appreciate your comprehensive feedbacks to me and other reviewers. Almost all my concerns have been addressed and I will increase my score to 7 from 6.
> > >
> > > Reply: Thank you for your reading and the kind reply! Your comments are helpful and we follow your comments to improve our paper.
> > >
> > > Q2: Just a couple of requests. If your paper ended up being published, please: Be cautious about using the term "adaptive".
> > >
> > > Reply: Thanks for your helpful suggestions! We follow your and other reviewers’ comments to improve our description and usage of the term “adaptive”. We agree with the reviewers that our method focuses on the adaptive data augmentation magnitudes on different parts of the input features related to informativeness. As we have written in the paper, we consider the informativeness of edges to a target node in terms of both time and topology when predicting the activities of the target node. If an edge is more close to the target node in terms of time or topology, it is more informative, as validated in [1]. In addition, this also meets the inductive bias of the recurrent neural network models [2], which process the temporal edges in the time order and conveys the messages between connected nodes across multiple hops. We provide theoretical analysis to support that our MeTA framework can produce adaptive data augmentation on the temporal graphs.
> > >
> > > On the other hand, we also agree with the reviewers, there are some methods in computer vision that adaptively select the data augmentation strategies to apply [5]. Note that the adaptive choosing of the data augmentation strategies is orthogonal to our method, which achieves adaptive data augmentation magnitudes on different parts of the input features for any data augmentation strategy on the temporal graph learning. As far as we know, ours is the first work that develops the data augmentation method for temporal graph learning. The adaptive choosing of the data augmentation strategies is worth exploration as a direction of future work and can further boost the performance of temporal learning combined with our methods.
> > >
> > > Overall, we follow your and other reviewers’ suggestions to emphasize the meaning of the term “adaptive” in our paper and discuss the future directions on adaptive data augmentation on temporal graph learning to improve our paper.
> > >
> > > Q3: Consider reporting MRR or Recall (or both) for all methods/datasets as I believe they are far more realistic metrics in this setting.
> > >
> > > Reply: Thanks for your helpful suggestions! Our evaluation metrics and experimental settings closely follow those of the most recent work [1,3] to ensure a fair comparison with the state-of-the-art methods. We agree with the reviewer that using additional metrics to evaluate our method like MRR [4], as the reviewer suggests, is beneficial for the more comprehensive evaluation. As we presented in our response to Q2 in your initial comments, evaluated by MRR [4], our MeTA achieves consistent and substantial improvements for TGN on all datasets and enhances TGN to outperform all the baseline methods, thanks to the adaptive data augmentation offered by our MeTA framework, which effectively augments the input features while preserving the essential semantic information.
> > >
> > > We follow your suggestions to add these experimental results in the paper to provide a more comprehensive evaluation.
> > >
> > >
> > > [1] Rossi, Emanuele, et al. "Temporal graph networks for deep learning on dynamic graphs." arXiv preprint arXiv:2006.10637 (2020).
> > >
> > > [2] Khandelwal, Urvashi, et al. "Sharp nearby, fuzzy far away: How neural language models use context." arXiv preprint arXiv:1805.04623 (2018).
> > >
> > > [3] Xu, Da, et al. "Inductive representation learning on temporal graphs." arXiv preprint arXiv:2002.07962 (2020).
> > >
> > > [4] Kumar, Srijan, Xikun Zhang, and Jure Leskovec. "Predicting dynamic embedding trajectory in temporal interaction networks." Proceedings of the 25th ACM SIGKDD International Conference on Knowledge Discovery & Data Mining. 2019.
> > >
> > > [5] Ekin D Cubuk, Barret Zoph, Dandelion Mane, Vijay Vasudevan, and Quoc V Le. Autoaugment: Learning augmentation policies from data. arXiv preprint arXiv:1805.09501, 2018.

---

### Official Review · Reviewer_WSXU · 2021-07-16

**Rating:** 7
**Confidence:** 4

**Summary:**

The paper introduces Memory Tower Augmentation (MeTA) --- an adaptive data augmentation approach for temporal graph networks (TGNs). The proposal aims to tackle overfitting and improve learning in TGNs. MeTA applies a hierarchical message-passing scheme to update TGNs' memory states. In addition, MeTA considers three different data augmentation strategies: i) perturb time, ii) remove edges, and iii) add edges with perturbed time.  The experiments show that MeTA achieves higher accuracy than existing TGNs (without data augmentation) on future link prediction and node classification tasks.


**Limitations And Societal Impact:**

The authors adequately address the limitations and the broader impacts of the proposal.

**Main Review:**

The proposed method (MeTA) is novel as it represents the first attempt to leverage adaptive data augmentation techniques for temporal graph networks. The results show that MeTA can boost the performance of the baselines. The main weakness is that MeTA scales linearly with the number of levels of memory modules ($L$). Also, I found section 3.2 particularly hard to follow, and the authors should consider rewriting it for clarity.

### Comments:
* Avoiding overfitting is one of the motivations (lines 3, 24). However, current TGNs are shallow models with 1 or 2 temporal aggregation layers. Is overfitting a problem in those methods? How much does the proposal alleviate it? Showing this would strengthen the paper.
*The experiment with PerturbTime can change the ordering of the events. Does it have any relevant implications? Do the authors have any intuition about why this ordering change does not negatively impact the performance?
* Figure 4 shows that "large values of p1 and small p2 results in poor performances". The authors explain that a large p1 increases the risks of breaking the semantic information, while a small p2 cannot augment the features sufficiently. Based on that intuition, why do large values for p1 (=0.5) and p2 (=1.0) still lead to good performance (Figure 4)?
*It would be valuable to report numbers for different values of $L>2$ in the supplementary material.
*For some TGNs, a reasonable choice is to sample/aggregate temporal information that respects time. In this approach, neighbors are events in the past (e.g., see CAWNet). Is that the case for the evaluated TGNs? If so, both topological and temporal distances would convey the same information.
*The authors should cite this related work: CAWNet (https://arxiv.org/abs/2101.05974).

### Minor (inconsequential) issues
* In Table 2, denoting MeTA (L=1) in boldface seems misleading since there is no reason for MeTA (L=1) to be faster than the original approach (TGN).
* Some typos: Gaussion (line 184), tmax (line 188), $s^i$ (line 218), "more earlier edges" (line 220), "magnitudes , as" (line 53), "information , as" (line 42).
* I don't find the comparison against the refeed method relevant (section 4.3, and line 241). It is clearly much slower than MeTA.

--------------------------------------
After the rebuttal phase, I have decided to increase my score to 7 (accept).


**Time Spent Reviewing:**

9

---

> ### Author Response · Authors · 2021-08-10
> **Response to Comments from Reviewer WSXU**
>
> We greatly thank the reviewer WSXU for his/her helpful and insightful comments. We provide our responses to the comments as follows.
>
> Q1: Avoiding overfitting is one of the motivations (lines 3, 24). However, current TGNs are shallow models with 1 or 2 temporal aggregation layers. Is overfitting a problem in those methods? How much does the proposal alleviate it? Showing this would strengthen the paper.
>
> Response: Thanks for your helpful suggestions! We observed “overfitting” of TGNs on the temporal graph learning in the experiments. For example, on the transductive edge prediction, TGN with and without our MeTA method achieves the following accuracy (%) on the training set and test set of MOOC, LastFM, Reddit, and Wikipedia datasets respectively:
>
> | Method|MOOC|LastFM|Reddit|Wikipedia|
> |-|-|-|-|-|
> | TGN [3] (on Training)|92.81 $\pm$ 0.6|88.30 $\pm$ 0.3|96.41 $\pm$ 0.3|94.79 $\pm$ 0.3|
> | TGN [3] (on Test)|81.38 $\pm$ 0.6|69.35 $\pm$ 0.2|92.56 $\pm$ 0.2|89.51 $\pm$ 0.4|
> | TGN + MeTA (on Training) |86.21 $\pm$ 0.3|77.08 $\pm$ 0.2|96.07 $\pm$ 0.2|93.62 $\pm$ 0.2|
> | TGN + MeTA (on Test)|83.84 $\pm$ 0.5|72.15 $\pm$ 0.2|94.19 $\pm$ 0.2|91.34 $\pm$ 0.3|
>
> We observe that there exist significant gaps between the performance of the baseline model TGN on the training sets and test sets of all datasets, where the training performance is much better than the testing performance. This implies the overfitting of TGN. In contrast, our MeTA method achieves substantial improvements for TGN on the test set and reduces the performance gap between training and testing for TGN trained with our MeTA. This validates that our MeTA method can mitigate the overfitting and improve the generalization of TGN. We follow your suggestions to add these experimental results in the paper, which helps to illustrate how our method alleviates overfitting.
>
> We agree with the reviewer that higher model complexity carries with it a higher risk of overfitting. Existing work finds that the graph neural networks (GNNs) on static graphs hold the best performance with a few layers [1], of which the overfitting is an issue and their generalization is improved by the data augmentation [2]. TGNs hold more complex model complexity than GNNs [3], of which the overfitting is observed. Therefore, in this work, we propose MeTA (Memory Tower Augmentation), an adaptive data augmentation (DA) approach for improving the temporal graph learning. We follow your suggestion to discuss this in detail to improve our paper.
>
> Q2: The experiment with PerturbTime can change the ordering of the events. Does it have any relevant implications? Do the authors have any intuition about why this ordering change does not negatively impact the performance?
>
> Response: Thanks for your questions! We agree with the reviewer that our time perturbation data augmentation can change the time ordering of two events especially when the time difference between the events is small. The time ordering change between two events, that are close in time, are widespread in the real world. For instance, a researcher can change the ordering of a lunch and a research meeting close to the lunch time depending on when his/her collaborator is available. Therefore, modeling these ordering changes can help enrich the training data and offers more supervision signals to enhances TGNs’ generalization. In addition, our MeTA, as we have written in line 118, applies low DA magnitudes to the informative parts of the input features and higher magnitudes to the less informative parts. Therefore, the time ordering changes are less likely to happen on the edges in the more informative parts, which can help to retain the semantic information and effectively augment the input features. We follow your suggestion to discuss this in detail to improve our paper.
>
> Q3: Figure 4 shows that "large values of p1 and small p2 results in poor performances". The authors explain that a large p1 increases the risks of breaking the semantic information, while a small p2 cannot augment the features sufficiently. Based on that intuition, why do large values for p1 (=0.5) and p2 (=1.0) still lead to good performance (Figure 4)?
>
> Response: Thanks for your question! In Fig. 4, the hyper-parameters $p_1=0.5$ and $p_2=1.0$ leads to better performance than the $p_1=0.5$ and $p_2=0.5$, since the former augments the less informative features more sufficiently with a higher p2. However, $p_1=0.5$ and $p_2=1.0$ leads to worse performance than the $p_1=0.1$ and $p_2=1.0$, since the latter reduces the risks of breaking the semantic information with a lower $p_1$. In Fig. 4, as your mentioned, the performance of TGN with MeTA is relatively smooth when parameters are within certain ranges (e.g., around $p_1=0.5$ and $p_2=1.0$), because our MeTA is robust to hyper-parameters on providing effective data augmentation for TGNs, as we have written on line 371. We follow your suggestion to discuss this in detail to improve our paper.
>
> Q4: It would be valuable to report numbers for different values of $L>2$ in the supplementary material.
>
> Response: Thanks for your helpful suggestions! We investigate the influence of the number of levels $L$ of MeTA on the performance. The node classification results of TGN with MeTA with different numbers of $L$ evaluated by the test ROC AUC (%) are presented in the following table:
>
> |Number of Levels $L$|MOOC|Reddit|Wikipedia|
> |-|-|-|-|
> |1|78.27 $\pm$ 0.7|67.72 $\pm$ 0.8|88.73 $\pm$ 0.3|
> |2|79.41 $\pm$ 0.8|68.37 $\pm$ 0.9|90.03 $\pm$ 0.3|
> |3|79.45 $\pm$ 0.9|68.42 $\pm$ 0.9|90.14 $\pm$ 0.4|
> |4|79.51 $\pm$ 0.8|68.45 $\pm$ 0.8|90.17 $\pm$ 0.3|
> |5|79.52 $\pm$ 0.9|68.46 $\pm$ 0.7|90.19 $\pm$ 0.3|
>
> Setting more levels with $L > 2$ leads to the incremental improvements on the TGNs’ performance compared with MeTA of $L = 2$. We find that $L = 2$ generally offers satisfactory performance, so we set the number of memory levels as $L = 2$ for all experiments, as we have written on lines 266 and 268. We follow your suggestion to discuss these results in detail to improve our paper.
>
> Q5: For some TGNs, a reasonable choice is to sample/aggregate temporal information that respects time. In this approach, neighbors are events in the past (e.g., see CAWNet). Is that the case for the evaluated TGNs? If so, both topological and temporal distances would convey the same information.
>
> Response: Thanks for your question! The baseline TGNs aggregate the temporal information in the past through the memory updates based on RNN, which is also utilized by CAWNet [4]. The temporal and topological distances are on different dimensions, of which the former is on time and the latter is on the topology. Both the temporal and topological distances, as you mentioned, are related to the edge's informativeness, so we design the data augmentation methods related to these distances to offer adaptive data augmentation. Please see our more detailed discussions on CAWNet in our response to Q6.
>
> Q6: The authors should cite this related work: CAWNet (https://arxiv.org/abs/2101.05974).
>
> Response: Thanks for your helpful suggestion! CAWNet [4] is a recently proposed TGN, which uses RNNs to encode the temporal information. We present the test accuracy (%) on inductive edge prediction of CAWNet as following:
>
> |Method|MOOC|LastFM|Reddit|Wikipedia|
> |-|-|-|-|-|
> |CAWNet [4]|80.22 $\pm$ 0.3|63.97 $\pm$ 0.3|91.78 $\pm$ 0.2|88.92 $\pm$ 0.3|
> |TGN [3]|80.73 $\pm$ 0.2|64.52 $\pm$ 0.3|91.62 $\pm$ 0.1|88.60 $\pm$ 0.2|
> | TGN + MeTA (Ours)|83.47 $\pm$ 0.2|67.22 $\pm$ 0.3|92.96 $\pm$ 0.1|90.82 $\pm$ 0.2|
>
>
> CAWNet achieves comparable performance to our baseline method TGN, while our MeTA method effectively enhances TGN to outperform CAWNet with significant improvements. We follow your suggestion to discuss CAWNet, and take CAWNet as an additional baseline for all experiments to improve our paper.
>
> Q7: In Table 2, denoting MeTA (L=1) in boldface seems misleading since there is no reason for MeTA (L=1) to be faster than the original approach (TGN).
>
> Response: Thanks for your question! We believe that the reviewer is referring to Table 5 instead of Table 2. As shown in Table 5, TGN with MeTA($L=1$) has the same number of memory updates per epoch as TGN in expectation. TGN with MeTA($L=1$) has close training time until convergence to the original TGN, where TGN with MeTA($L=1$) is slightly faster than original TGN, since TGN with MeTA ($L=1$) empirically needs fewer epochs to converge than TGN. We follow your suggestion to discuss this to improve our paper.
>
> Q8: Some typos: Gaussion (line 184), tmax (line 188),  (line 218), "more earlier edges" (line 220), "magnitudes , as" (line 53), "information , as" (line 42).
>
> Response: Thanks for your helpful suggestions! We follow your suggestions to revise our paper.
>
> Q9:  I don't find the comparison against the refeed method relevant (section 4.3, and line 241). It is clearly much slower than MeTA.
>
> Response: Thanks for your questions! We have provided the experimental results on comparison with refeed in Table 5 (row 2), and explain the results from line 318 to 337. As you mentioned, the refeed method is slower than MeTA empirically and theoretically.
>
> Q10: The main weakness is that MeTA scales linearly with the number of levels of memory modules ($L$). Also, I found section 3.2 particularly hard to follow, and the authors should consider rewriting it for clarity.
>
> Response: Thanks for your helpful suggestions! MeTA scales linearly with the number of levels, but just as shown in our response to Q4 and Section 3 and 4, $L$ is a small constant value invariant to the input data. The level number $L = 2$ generally offers satisfactory performance, and $L > 2$ offers negligible improvements than $L = 2$. Therefore, increasing $L$ to $L>2$ is unnecessary and MeTA’s linear scaling with $L$ is thus not an issue. We follow your suggestions to emphasize this and rewrite Section 3.2 to improve our paper.

---

> > ### Author Response · Authors · 2021-08-10
> > **References**
> >
> > [1] Kipf, Thomas N., and Max Welling. "Semi-supervised classification with graph convolutional networks." arXiv preprint arXiv:1609.02907 (2016).
> >
> > [2]  Zhao, Tong, et al. "Data augmentation for graph neural networks." arXiv preprint arXiv:2006.06830 (2020).
> >
> > [3] Rossi, Emanuele, et al. "Temporal graph networks for deep learning on dynamic graphs." arXiv preprint arXiv:2006.10637 (2020).
> >
> > [4] Wang, Yanbang, et al. "Inductive Representation Learning in Temporal Networks via Causal Anonymous Walks." arXiv preprint arXiv:2101.05974 (2021).

---

> > ### Comment · Reviewer_WSXU · 2021-08-26
> > **Thanks for the feedback**
> >
> > I would like to thank the authors for their feedback, which addressed most of my concerns, especially Q1, Q2, Q4, Q7-9 (minor), and Q10.
> >
> > Regarding Q6, I took the time to run CAWNet and got accuracies (not AP) around 98% (Reddit) and 99% (Wiki) in the transductive setting, which is much better than TGN+MeTA. Overall, I have found CAWNet to work better than TGN for datasets without edge features (e.g., LastFM). Therefore, the results in the answer for Q6 seem a little odd to me. If the authors decide to include a comparison against CAWNet in the paper, I would recommend first trying to reproduce the results in the original paper to make sure the reported accuracies are correct.

---

> > > ### Author Response · Authors · 2021-08-29
> > > **Thanks and Response to New Comments from Reviewer WSXU**
> > >
> > > Thanks for your time and responses! We provide our responses to the new comments as follows.
> > >
> > > Q1. I would like to thank the authors for their feedback, which addressed most of my concerns.
> > >
> > > Reply: Thank you for your check and kind reply!
> > >
> > > Q2. Regarding Q6, I took the time to run CAWNet and got accuracies (not AP) around 98% (Reddit) and 99% (Wiki) in the transductive setting, which is much better than TGN+MeTA.
> > >
> > > Reply: Thanks for your questions! We appreciate the efforts that you take on the experiments, which is beneficial for comparing the details of different implementations. We would like to explain our experiments in detail to address your concerns.
> > >
> > > First, to ensure that our implementation on CAWNet is correct, we have followed the official implementation of CAWNet provided in:
> > >
> > > https://github.com/snap-stanford/CAW
> > >
> > > to implement the model of CAWNet and report its performance in our reply.
> > >
> > > You observed some differences on the test accuracy between our reported results and your running results. We analyze your observations as following. All of our experimental results are given by the evaluation protocol provided in the official implementation of the paper [1] in:
> > >
> > > https://github.com/StatsDLMathsRecomSys/Inductive-representation-learning-on-temporal-graphs,
> > >
> > > which is also widely utilized in other work such as [2]. We strictly fix the evaluation protocol for all the models as this implementation to ensure a fair comparison.
> > >
> > > On the other hand, in the CAWNet's official implementation, its authors have a different evaluation protocol. We carefully check the evaluation protocol of CAWNet and [1], and find that there exist differences between them, e.g., on choosing negative samples. These differences make the same model present different accuracy results evaluated on CAWNet’s official code and [1]. We observe that the CAWNet’s evaluation protocol generally returns higher accuracy for different models than the evaluation protocol of [1], which can account for the gaps that you observed from different implementations. Note that CAWNet is the most recently proposed model on the temporal graph learning, for which we did not use its evaluation code initially to ensure a fair comparison with existing works. Note that the CAWNet’s paper does not report the accuracy results.
> > >
> > > For a more comprehensive evaluation, we additionally use the evaluation protocol of CAWNet’s official implementation to compare the performance of different methods. We find that taking the CAWNet's evaluation protocol to evaluate the models, TGN with our MeTA method is able to outperform the baselines including CAWNet on all datasets. Moreover, as the data augmentation methods regularizing the temporal graph networks for better generalization, we observe that our methods are also beneficial for CAWNet’s effectiveness when applied to its training.
> > >
> > > We follow your suggestions to discuss these experimental details to improve our paper. We will release the code for the implementation of our paper, where we provide both evaluation protocols from the CAWNet’s paper and [1], and emphasize that we follow the existing evaluation protocol in [1] for a fair comparison.
> > >
> > > [1] Xu, Da, et al. "Inductive representation learning on temporal graphs." arXiv preprint arXiv:2002.07962 (2020).
> > >
> > > [2] Rossi, Emanuele, et al. "Temporal graph networks for deep learning on dynamic graphs." arXiv preprint arXiv:2006.10637 (2020).

---

### Decision · Program_Chairs · 2021-09-27

**Decision:**

Accept (Poster)

**Comment:**

After discussions between the reviewers and authors it seems as though a consensus has been reached amongst 3 of the reviewers that this paper is clearly worthy of acceptance.  The reviewer on the side of rejection has not engaged in discussions with the authors or in internal discussions on the paper.  The review against acceptance also lacks details in the objections raised to the paper.

I want to thank the authors for their detailed responses and for listening to and incorporating the feedback from the reviewers and thank those reviewers who did for engaging in discussions with the authors.